# Phase diagrams of pharmaceutical solvates from mechanochemistry

Fragkoulis Theodosiou [1], Toby J. Blundell [1], John S. O. Evans [1], Patricia Basford [2,3], Noalle Fellah [4] & Aurora J. Cruz Cabeza [1] ✉

Crystalline solvates, including hydrates, hold untapped potential in pharmaceutical development, yet their exploitation remains minimal due to the difficult and laborious task of unequivocally establishing their physical stabilities. We introduce Controlled Solvent-Activity Liquid-Assisted Grinding (CSA-LAG), a mechanochemical protocol that offers solvate phase boundary elucidation by varying the activity of a chosen solvent in defined binary/ternary mixtures and analysing the equilibrated resulting solid form. Using small API amounts, CSA-LAG reaches equilibrium within minutes and yields critical solvent activities that delimit neat, hydrated, solvated and competing-solvate domains. The method uses mixtures of known thermodynamic activities, requires far less material and time than traditional slurries and affords high reproducibility. Applied to four pharmaceutical compounds, CSA-LAG reproduces slurry boundaries and quantifies activity thresholds for single, stepwise and competitive solvations. Defining these boundaries enables rational form selection and process design either by avoiding or targeting solvates, whilst turning a month-scale empirical screening into a rapid, thermodynamically guided workflow.

During drug manufacturing, active pharmaceutical ingredients (APIs) are exposed to an array of different solvents, antisolvents and vapours under a range of temperature and pressure conditions[1,2]. Such exposure can lead to undesired solid form transformations, mainly solvation or desolvation, negatively impacting the attributes of APIs (sometimes even as a function of batches)[3,4]. Perhaps driven by their reduced ability to pack efficiently in the solid state[4], large flexible compounds are particularly prone to forming hydrates and solvates in the solid state[5,6]. Given that new drug compounds are becoming increasingly complex[5], crystalline solvates are becoming more common in drug delivery. For example widely used drugs such as sildenafil citrate[7], doxycycline[8], azithromycin[9] or amoxicillin[10], are commercialised as hydrates and other types of non-aqueous solvates have recently been used to formulate seven new marketed drugs[11]. Understanding the thermodynamic and kinetic stabilities of pharmaceutical solvates and hydrates is key for their production, packaging and storage[1,12].

Phase diagrams reveal the regions of thermodynamic stability of solid forms, crucially solvated and unsolvated forms, as a function of environmental conditions. They provide engineers with the necessary information to design and optimise processes for the manufacture of the desired solid forms and prevent unwanted conversions. The experimental elucidation of phase diagrams, however, can be tedious and time consuming with 'slurry experiments' providing the gold-standard methodology. In slurry experiments, excess solids are stirred in solutions of known solvent activity and their structure is monitored as a function of time. Given infinite time, slurry experiments should lead to the most stable solid forms under the chosen environmental conditions[13]. Ascertaining whether equilibrium has been reached in a slurry experiment, however, can be challenging since kinetically hindered conversions often require weeks to months of equilibration time[3,14–16]. Systems having fast hydration kinetics (e.g. non-stoichiometric hydrates) can also be studied with other methods such as dynamic vapour sorption, gravimetric monitoring of solids

[1]Department of Chemistry, Durham University, Durham, UK. [2]Material Sciences, Pharmaceutical R&D, Pfizer Central Research, Kent, UK. [3]Particology Ltd., Sandwich, Kent, UK. [4]Chemical Research and Development, Pfizer Inc., Groton, CT, USA. ✉e-mail: aurora.j.cruz-cabeza@durham.ac.uk

stored over saturated salt solutions or differential scanning calorimetry (DSC)[17–23].

Mechanochemistry[24] has recently emerged as an environmentally friendly technology used in drug research, development[25–27] and manufacturing[28]. As a tool for solid form screening liquid-assisted grinding (LAG) has been shown to rapidly lead to new polymorphs of complex pharmaceuticals, as well as the facile formation of multi-component crystals including hydrates and solvates[26,29–33]. Over the last 23 years or so efforts have been made to understand mechanochemical reaction pathways[34–36] and rationalise LAG solid form outcomes as a function of the liquid nature, composition and amount. In 2002 Shan et al. first reported that the formation of cocrystals via ball-mill grinding could be significantly accelerated by adding 'minor amounts of an appropriate solvent' to the mill, introducing LAG[37]. Early works with LAG were referred to as 'solvent-drop grinding' given that experimentally, only a few drops of solvent were added to the milling experiments[38]. In 2004 it was shown that LAG with different solvents can lead to different polymorphs[39] and in 2016 that those outcomes are a consequence of crystal size effects[32]. The fundamental role of the solvent in LAG interconversion reactions is still not fully understood. Recent population balance modelling work has shown that crystal breakage, crystal growth and Oswald ripening processes occur simultaneously in the mill and that the LAG solvent can impact this delicate balance and thus, the LAG outcomes[26]. Crucially, the solvent's nature and amount (relative to solids) impact the rate of crystal growth of the new phases in the mill, with greater solvent volumes and higher solubilities promoting larger crystal formation[26,40].

In the early 2000s the Jones group conducted several studies demonstrating that pharmaceutical hydrates[41] and solvates[42] could be generated using simple LAG. The resulting LAG phases were predicted computationally through lattice energy calculations[33], with entropic[43] and conformational effects being incorporated into such models recently[44]. In 2011, Friščić et al. reported that LAG with alcohol:water solvent mixtures could lead to different hydrate forms depending on the composition of the mixture used[31]. Whilst the authors noticed that the relative water composition of the mixtures should impact the effective water activities in the LAG experiments, variable activities were never implicitly considered in these early (or in later) works on inorganic hydrates[45], solvates[29] and competing solvate studies[30,46,47].

Here we present the framework of thermodynamic equations that rationalise solvate formation in the mill and develop a robust experimental method for LAG under controlled-solvent activity conditions (CSA-LAG). The central innovation of CSA-LAG lies in using solvent mixtures of defined composition to systematically and reproducibly tune the bulk solvent activity in the milling environment, enabling activity-based mapping of solvate/hydrate boundaries. The solvent mixtures consist of at least one active solvent (able to solvate the system in the solid-state) and a passive solvent, also referred to as the carrier solvent (used to tune the composition of the mixture and the activity of the active solvent). CSA-LAG allows for the accurate and rapid exploration of phase diagrams of solvates and hydrates, including systems with multiple stoichiometries and competing solvates. We show the robustness, accuracy and generality of CSA-LAG by exploring multiple solvation landscapes for four compounds of pharmaceutical interest at room temperature.

## Results and discussion
### Thermodynamic framework
Solvate formation can be interpreted within the broader framework of phase equilibria, wherein the relative stability of solid forms is governed by the chemical potential[48]. At thermodynamic equilibrium, the chemical potential of each component remains uniform across all coexisting solid and liquid phases. Deviations from equilibrium introduce a driving force for solvation or desolvation, determined by the solvent activity in the liquid phase[49]. Within the context of CSA-LAG,

these thermodynamic principles can be adapted to describe relevant mechanochemical transformations, enabling the expression of solvate equilibria in terms of equilibrium constants, critical activities and changes in free energy. In our framework nomenclature we use A for the API, S and T for different solvents, $m$, $n$ and $w$ for different solvent to API stoichiometries in the solid-state, (s) and (l) to indicate the solid and liquid states, $K$ for the equilibrium constant, $Q$ for the reaction quotient and α for the activities. For simplicity, the stoichiometry of the API in the solid-state is considered as 1 relative to $m$, $n$ and $w$; inclusion of a different stoichiometry for the API in the formulations is straightforward.

From this foundation three generalised scenarios emerge that capture the most common solid-state solvation phenomena: (a) single solvation, (b) stepwise solvation and (c) competing solvation. These scenarios are illustrated in Fig. 1 and serve as the theoretical basis for the subsequent analysis. Scenario (a), the most prevalent, involves the transformation of a neat API into a single solvate of defined stoichiometry. Scenario (b), while less common, is frequently encountered in hydrate systems, where the API may exist in neat form and as different solvate structures with the same solvent but different stoichiometries. Scenario (c), which is comparatively rare, addresses the thermodynamic competition between two different types of solvates of the same API. Although additional scenarios can be envisaged—such as competition among three solvents for solid-state solvation, multi-step solvation involving three or more stoichiometries, or the formation of heterosolvates—these are significantly less common so they will not be presented and can be derived with analogous thermodynamic formulations as those established here.

The derivation of the equilibrium constants in Fig. 1 is straightforward after equating the activities of all solids to one. At equilibrium, the activity of the solvent in the reactions is known as the critical solvent activity. Critical solvent activities in Fig. 1 are identified with the aid of an asterisk superscript with the specific solvate reaction it refers to given in brackets. For example in scenario (a) where a single solvate forms, $\alpha_S^*\{AS_m\}$ is the critical solvent activity of solvent S in the equilibrium with respect to the formation of the $AS_m$ solvate. The actual environmental solvent activity of the experiment (i.e. the activity conditions of the active solvent added to the LAG experiment) is $\alpha_S$. The evolution of the reaction can be predicted with the aid of the change of free energy equation[48] which requires knowledge of the equilibrium constant of the reaction (thus the critical activities, $\alpha_S^*$) and the reaction quotient (the actual environmental solvent activities, $\alpha_S$). When $K > Q$ the reaction evolves towards the products, when $K < Q$ the reaction evolves towards the reactants, and at $K = Q$ the reaction is at equilibrium.

In scenario (a) the API $A(s)$ is milled with a liquid containing S with the possibility of the solvate $AS_m(s)$ forming. The critical solvent activity $\alpha_S^*\{AS_m\}$ governs the equilibrium of the reaction. If $\alpha_S > \alpha_S^*\{AS_m\}$ the ratio between the actual and critical activities ($\alpha_S/\alpha_S^*\{AS_m\}$) is greater than 1 and thus $\triangle G$ is negative and the system will evolve towards the products (the formation of the solvate). If $\alpha_S < \alpha_S^*\{AS_m\}$ solvation does not occur and if $\alpha_S \cong \alpha_S^*\{AS_m\}$, a mixture of the solvated and unsolvated solid A should be obtained.

In scenario (b) $A(s)$ can be solvated with the same solvent S but leading to two solvates of different stoichiometries $AS_m(s)$ and $AS_n(s)$ where $n > m$. In this scenario, the equilibria are defined by two critical solvent activities $\alpha_S^*\{AS_m\}$ and $\alpha_S^*\{AS_n\}$. Since n > m, $\alpha_S^*\{AS_n\} > \alpha_S^*\{AS_m\}$. The evolution of the system will be dictated again by the actual solvent activity in the experiment $\alpha_S$ with the expected outcome being predictable according to $\triangle G$. If $\alpha_S < \alpha_S^*\{AS_m\}$, there will be no solvation; if $\alpha_S^*\{AS_m\} < \alpha_S < \alpha_S^*\{AS_n\}$, solvate $AS_m$ forms; if $\alpha_S^*\{AS_n\}<\alpha_S$, $AS_n$ forms and at conditions close to the critical activities mixtures can form.

In scenario (c) an API can form two different solvates with solvents S and T with stoichiometries $m$ and $w$, respectively. If only solvent S or solvent T is present, the equations in Fig. 1a apply and the evolution of

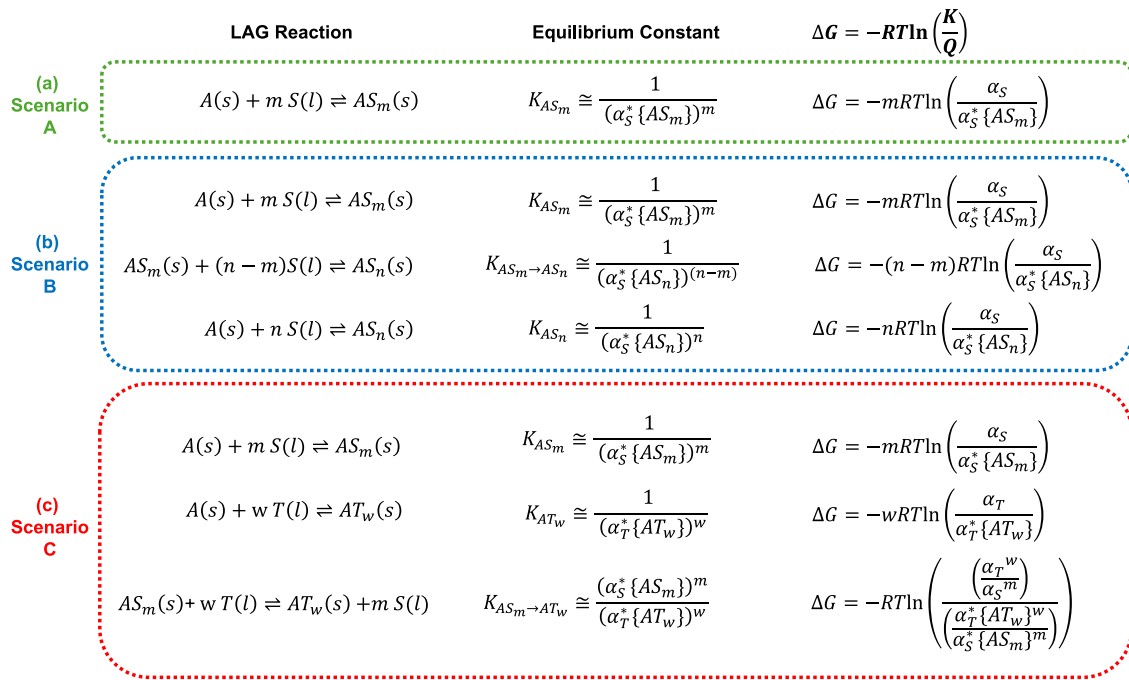

**Fig. 1 | Thermodynamic relationships for LAG solvation reactions.** LAG reactions, equilibrium constants, and changes in the reactions' free energies for three solvation scenarios: **a** single solvation, **b** stepwise solvation, and **c** competing solvation. A is used for the API, S and T for two different solvents and $m$, $n$ and $w$ for the stoichiometries. (s) and (l) refer to the solid and liquid states, respectively. Activities ($\alpha$) at equilibrium are indicated with an * superscript.

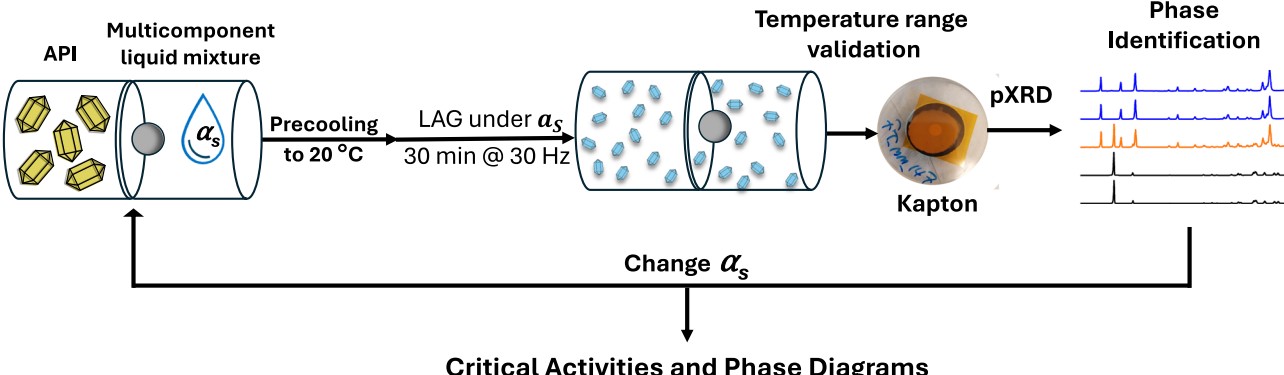

**Fig. 2 | Workflow for CSA-LAG phase diagram mapping.** Schematic representation of our CSA-LAG methodology describing the iterative approach for the exploration of the physical stabilities of APIs under controlled solvent activity environments by varying the active solvent activity ($\alpha_s$) used during milling.

the reaction is dictated by the activity of the active solvent. The last reaction in Fig. 1c shows the case of competing solvation. If both active solvents are present in the mill and both solvent activities lie above the critical activities to produce both solvates, $\alpha_S^*\{AS_m\}$ and $\alpha_T^*\{AT_w\}$, then there will be a case of competing solvation. In this case, the evolution of the reaction is governed by the ratio of the activities of the solvents. To simplify the discussion and given that the competing systems studied here are monosolvates we equate $m$ and $w$ to 1. The ratio of the activities of the two active solvents $\alpha_T/\alpha_S$ indicates the evolution of the system. If $\alpha_T/\alpha_S > \alpha_T^*\{AT\}/\alpha_S^*\{AS\}$, solvate AT forms; if $\alpha_T/\alpha_S < \alpha_T^*\{AT\}/\alpha_S^*\{AS\}$, solvate AS forms; at the boundary a mixture of AT and AS forms. If the individual activities are below both critical activities, solvation will not occur.

Given this thermodynamic framework the key challenge experimentally is precisely controlling the solvent activities in the mill to enable the determination of the critical activity values of the different reactions. This can be rapidly and reproducibly achieved with our introduced CSA-LAG methodology.

## CSA-LAG methodology

CSA-LAG leverages mechanochemistry to enable fast, robust and small-scale elucidation of solvate phase diagrams through the determination of critical solvent activities (Fig. 2). To achieve this, we carefully control the environmental milling conditions by using liquid mixtures of solvents with finely selected compositions and solvent activities. Except for two systems (which will be discussed later), we employ a liquid-to-solids ratio of 2 μL/mg with loadings of 100 mg of solids and 200 μL of liquid mixture for all CSA-LAG experiments. Given the ubiquitous presence of atmospheric water, careful handling of the milling jars and powders is essential to prevent moisture uptake. To mitigate for this, jars and crystalline powders were dried in an oven for 2 h prior to use and liquid mixtures were freshly prepared from sealed solvent bottles immediately before each experiment. After loading the jars with the solid and liquid samples, a single milling ball was added and the jars were sealed, precooled to 20 °C and shaken for 30 min at 30 Hz using a Retsch MM400 mill. Immediately after milling, the jars were removed from the mixer-mill (to prevent heat

accumulation from motor heat dissipation), opened and the milled solids were transferred to a PXRD sample holder. Samples were then immediately sealed with polyimide (Kapton) tape and analysed via PXRD without any delay. The use of Kapton is critical, as it isolates the sample from ambient conditions, preventing solvent loss or water uptake. The Kapton adds a 'halo' to the resulting PXRD pattern which can be simply subtracted as background (Supplementary Information Section 1.8.1). PXRD analysis was used to determine the resulting solid forms (Supplementary Information Section 2.3). Experiments were then repeated across a range of liquid mixture compositions and solvent activities ($\alpha_S$), enabling rapid mapping of solid form outcomes as a function of solvent activity. These variations reveal the critical activities associated with different equilibria, thereby delineating phase boundaries in the solvate phase diagram.

A key aspect of CSA-LAG is the preparation of liquid mixtures with finely controlled compositions and solvent activities (Supplementary Information Sections 1.6 and 1.7). These mixtures must be prepared using high-purity solvents, dried over molecular sieves and fully miscible across all ratios. Solvents with low volatility are preferred, as even minor compositional changes (e.g. due to evaporation) can significantly alter the thermodynamic properties of the liquid mixtures. To design the mixtures, we first identify the active solvent (i.e. the solvent responsible for solid-state solvation) and an inactive carrier solvent. Binary mixtures are then prepared to span solvent activities from 0 to 1 in increments of 0.1 (Supplementary Information Sections 1.6 and 1.7).

The required compositions of those mixtures are calculated using binary vapour-liquid equilibrium (VLE) data at 25 °C, with solvent activity as a function of molar fraction determined using ASPEN Plus v14.1, via the non-random two-liquid model (NRTL). Importantly, these activity calculations consider only solvent-solvent equilibria, excluding the influence of dissolved API. This is a pragmatic approximation widely adopted across pharmaceutical crystallisation and hydrate studies for non-electrolyte systems[50]. Under these conditions, the bulk solvent activity (from validated VLE models) serves as an adequate control parameter. Consistent with prior hydrate literature[1], we adopt the same practical approximation for CSA-LAG, while explicitly acknowledging potential limitations. Experimental validation[16,51], has shown, at saturation, dissolved API concentrations are typically low and exert minimal influence on solvent activity—quantified at less than 6% in representative systems[16,51]—well below our measured activity increments ($\Delta\alpha = 0.1$) and within typical VLE model uncertainties ($\pm 1$–5%)[52].

Industrial crystallisation protocols routinely rely on NRTL-derived bulk water activities to control an/hydrate equilibria in solution crystallisation. While this guidance does not explicitly address mechanochemical (LAG) environments, our results indicate that the bulk solvent activity remains an informative and historically overlooked control variable for mechanochemistry when the API is present predominantly as a solid and its dissolved fraction is small. We therefore use NRTL-calculated activities as a practical activity scale here, while explicitly delineating the limits of this approximation and outlining future validation experiments in Section 'CSA-LAG: methodological considerations'[50].

CSA-LAG enables activity-controlled mapping of solvation landscapes beyond water. Water activity can be probed directly (e.g. calibrated RH meter), whereas non-aqueous solvent activities are experimentally determined from vapour-pressure/VLE measurements or headspace analysis; here we use NRTL-derived bulk activities from curated VLE data as a practical, transferable scale. Because hydrates have historically dominated crystallisation form screenings owing to their ubiquity and regulatory relevance, activity-based mapping of non-aqueous solvates is far less common. CSA-LAG provides a unified, reproducible protocol to determine solvation boundaries across diverse solvent systems.

In CSA-LAG solids loading is a tunable parameter that can be adjusted alongside liquid volume, milling times and the number of milling balls. Provided the reactions reach equilibrium the outcomes are governed by thermodynamics and remain independent of the absolute scale. To accommodate different experimental needs, increasing the solid quantity requires a proportional increase in liquid (we use an overall 2 µL/mg ratio) and may also necessitate longer milling times or additional balls to ensure equilibrium is achieved. An initial exploration of the variables is recommended if the loadings are to be altered prior to map generation.

Temperature is another critical factor in these equilibria. Our current milling setup does not allow for active temperature control. To address this jars and milling balls were precooled to 20 °C prior to milling. We observed that milling raises the jar temperature by approximately 5 °C (measured ex situ using an infra-red thermometer), hence we achieve an effective milling temperature of ~25 °C after precooling to 20 °C (Supplementary Information, Section 2.4)—for consistency with the temperature of the VLE data used for activity calculations. Future iterations of the setup will incorporate more precise temperature control. In the absence of such control, we recommend monitoring temperature together with appropriate precooling.

## Systems

In this section we introduce the APIs and solvates studied and give examples of the solvent mixtures and their activities as a function of composition. Details for all solvent mixtures used are given in the Supplementary Information (Sections 1.6 and 1.7). We use the following naming convention: (i) non-capital letter abbreviations are used for compounds (including solvents), (ii) capital letter abbreviations are used for crystal forms followed by a Roman numeral or Greek letter indicating the identity of the polymorph and (iii) subscripts are used for stoichiometries. The stoichiometry of hydrates is also indicated with an abbreviation with HH referring to the hemihydrate, MH to the monohydrate and DH to the dihydrate systems. For example, 4OHBZM·HH-I refers to 4-hydroxybenzamide hemihydrate form I.

Since we wanted to show the generality and universality of CSA-LAG, four APIs of various molecular complexities were selected for our experimental studies (Fig. 3a), namely theophylline (theo), 4-hydroxybenzamide (4ohbzm), carbamazepine (cbz) and nitrofurantoin (nf). These four systems cover from small rigid (theo) or medium partially rigid compounds (cbz) to small lightly flexible (4ohbzm) and larger moderately flexible compounds (nf). The systems were also chosen because they are all known to form hydrates of diverse stoichiometries (which we wanted to explore) as well as a variety of other solvates of various nature (Table 1). The diverse nature and solid form richness of these APIs, therefore, make them ideal to explore CSA-LAG across a diverse range of molecular and solvate systems. We list all crystal forms studied for these four APIs in Table 1 including their abbreviations, compositions and corresponding Cambridge Structural Database refcodes.

To explore the equilibria between the relevant forms shown in Table 1, we generated a variety of binary and ternary solvent mixtures for the CSA-LAG using solvents such as water (h₂o), ethanol (etoh), acetone (ace), dimethylformamide (dmf), dimethylsulphoxide (dmso) and ethylene glycol (eg). The choice of solvents for the generation of the mixtures was made based on two distinct conditions: (a) one solvent requires to be active (e.g. leading to hydration/solvation) and (b) a second solvent is required to modulate the activity of the first through compositional variations (hence full miscibility with the active solvent is required) and, for simplicity, this second solvent is typically chosen to be inactive. The choice of the active solvent is dictated by the hydrate or solvate of interest. We studied four diverse active solvents, namely water, dmso, dmf and acetone (Table 1). The choice of inactive solvent was informed by the literature: it is simply a solvent leading to crystals of the unsolvated API under study. Common passive solvents

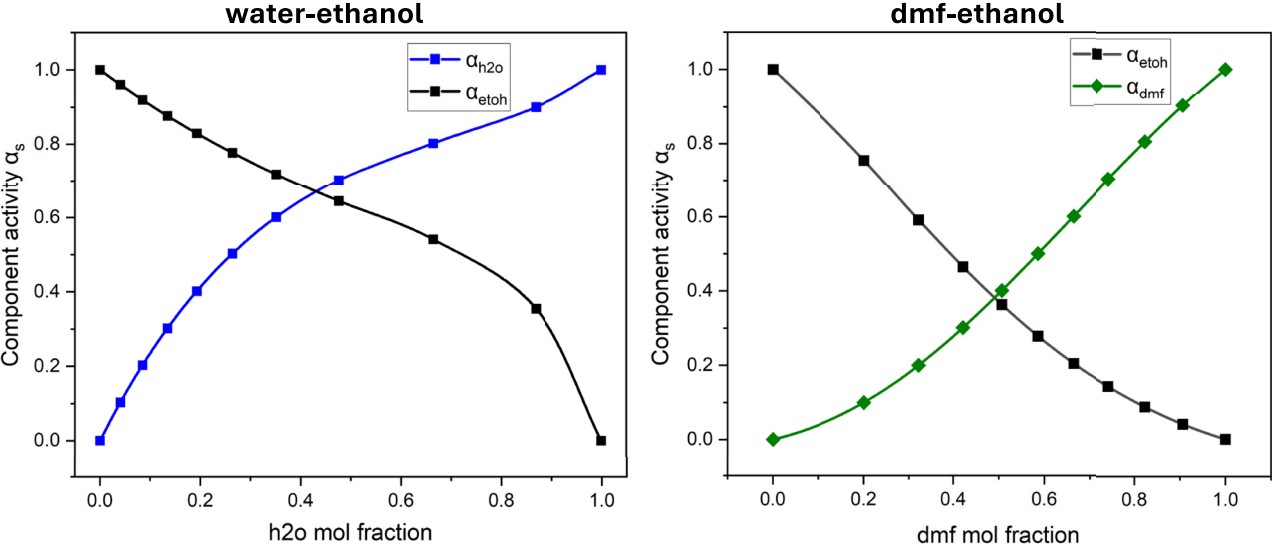

**Fig. 3 | Pharmaceutical compounds and binary solvent system behaviours. a** Molecular structures of APIs explored in this work, namely theophylline, 4-hydroxybenzamide, carbamazepine and nitrofurantoin. **b** Examples of binary mixtures used in this work and their solvent activity dependence with composition.

**Table 1 | Summary of the systems, crystal forms, CSD-refcodes, crystal form compositions, polymorphs, and liquid mixtures used in this study**

| System | Crystal form | CSD-refcode | Crystal composition | CSA-LAG/slurries liquid mixture (active: carrier)[a] |
|---|---|---|---|---|
| Theophylline (theo) | THEO-II<br>THEO-MH-I<br>THEO-DMSO-I | BAPLOT01<br>THEOPH05<br>RIGYEM | theo<br>theo:$h_2o$<br>theo:dmso | –<br>**$h_2o$**:ace<br>**dmso**:etoh |
| 4-hydroxybenzamide (4ohbzm) | 4OHBZM-I<br>4OHBZM-HH-I<br>4OHBZM-MH-I<br>4OHBZM-ACET-I | VIDMAX<br>GESCOY<br>JIXCOI01<br>2443101[b] | 4ohbzm<br>4ohbzm:$(h_2o)_{0.5}$<br>4ohbzm: $h_2o$<br>4ohbzm:ace | –<br>**$h_2o$**:eg<br>**$h_2o$**:eg<br>**ace**:eg |
| Carbamazepine (cbz) | CBZ-III<br>CBZ-DH-I<br>CBZ-DMSO-I | CBMZPN02<br>FEFNOT<br>UNEYIVO1 | cbz<br>cbz:$(h_2o)_2$<br>cbz:dmso | –<br>**$h_2o$**:etoh<br>**dmso**:etoh |
| Nitrofurantoin (nf) | NF-β<br>NF-MH-II<br>NF-DMF-II | LABJON<br>HAXBUD<br>2443102[b] | nf<br>nf:$h_2o$<br>nf:dmf | –<br>**$h_2o$**:eg<br>**dmf**:etoh |

[a]Liquid mixtures used for CSA-LAG in scenarios (a) and (b) and slurries, active solvents are represented in bold. Compositions were explored for mixtures with activities from 0 to 1 in steps of 0.1.
[b]CSD deposition numbers for the new solvates identified in this work.

used in crystallisation are ethanol, methanol, and acetone. Alternatively, passive solvents can be explored by LAG experiments. Ethylene glycol was also selected since it is usually passive and has a high boiling point—thus solvent evaporation is minimised. All binary mixtures used for each system are given in Table 1, details of their generation in the Supplementary Information (Sections 1.6 and 1.7) and activity against composition behaviour shown for two of the mixtures in Fig. 3b (**$h_2o$**:etoh and the **dmf**:etoh). The dependence of the activity

on molar composition in the mixtures is highly system-dependent and can significantly deviate from linearity and Raoult's law.

**Application to single solvation (scenario A)**
In this application of CSA-LAG we explore the mapping of a single solvate system per case study. In Fig. 4 the resulting PXRDs of the milled powders right after CSA-LAG are shown for two case studies: the formation of THEO-MH-I and NF-DMF-II. For the first case study, the

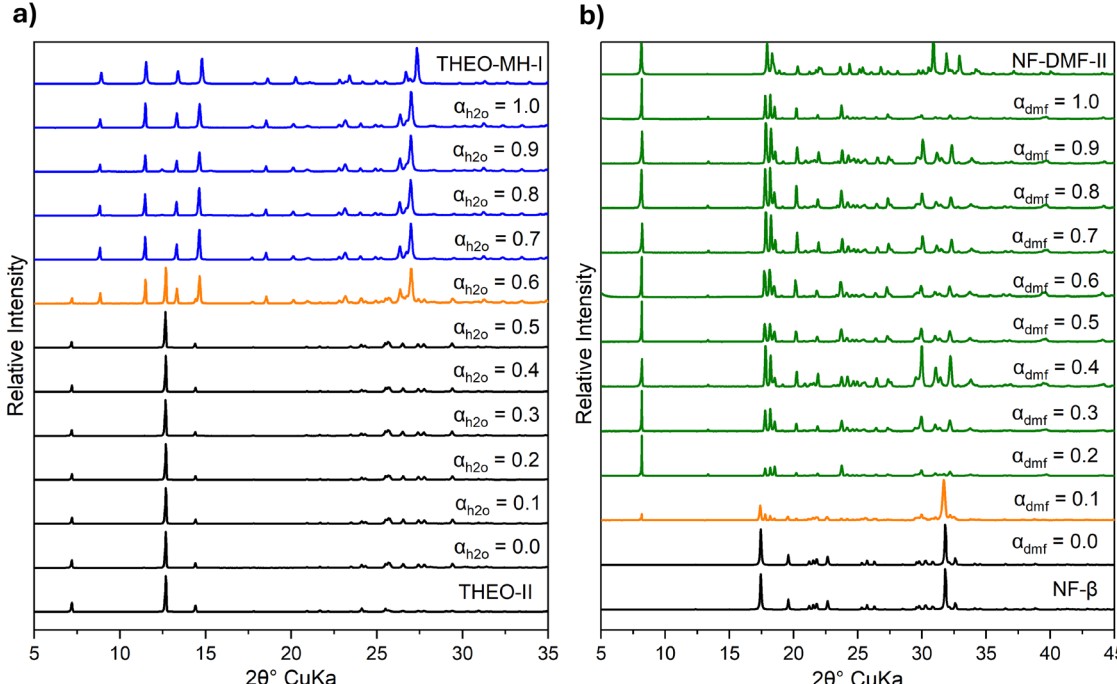

**Fig. 4 | Phase identification from CSA-LAG experiments using powder X-ray diffraction.** PXRD patterns of milled products of CSA-LAG at $25 \pm 1\,°C$ as a function of solvent activity for (**a**) theo with $h_2o$:ace mixtures and **b** nf with dmf:etoh mixtures. Patterns are coloured black for unsolvated forms, blue for hydrates and green for solvates. Mixtures of solid phases are presented in orange. The phases are identified by comparison of the resulting patterns to simulated PXRD diffractograms from single crystal structures. The theo mixture at 0.6 and nf mixture at 0.1 were further refined by Rietveld refinement.

equilibrium explored is THEO-II + $H_2O$ ⇌ THEO-MH-I. Here the solvent mixture employed contains water (the active solvent) and acetone (carrier solvent)[16]. For the second case study the equilibrium explored is NF-β + DMF ⇌ NF-DMF-II. Here, the solvent mixture contains dmf (active solvent) and ethanol (carrier solvent). In both case studies, the unsolvated forms of theo and nf are obtained at $\alpha_S = 0$. As the active solvent activities are increased, an activity is found where conversion to the solvates is observed, as seen by the appearance of Bragg peaks corresponding to the anticipated solvate forms (Fig. 4). The activity leading to the observation of new diffraction peaks together with the prior lower one explored, gives a range where the critical activity lies. For theo, new peaks are observed upon milling at $\alpha_{h2o} = 0.6$ whilst for nf at $\alpha_{dmf} = 0.1$, leading to ranges containing the critical activities of 0.5–0.6 and 0.0–0.1 respectively. The ranges containing the critical activities obtained by CSA-LAG correlate very well with those established by slurry experiments, of 0.56–0.61[14] and 0.07 – 0.10 for the THEO-MH-I and NF-DMF-II equilibria, respectively. In both these examples, the activities leading to new diffraction peaks contained a mixture of forms (THEO-MH-I and THEO-II at $\alpha_{h2o} = 0.6$ and NF-DMF-II and NF-β at $\alpha_{dmf} = 0.1$) which were confirmed and quantified by Rietveld refinement (Supplementary Information, Section 1.8.3). The critical activity ranges obtained with our method can be further explored by performing CSA-LAG at smaller activity increments around the boundary. We note, however, that the preparation of the required mixtures can become difficult for some systems, especially if the $\alpha_S$ is highly sensitive to compositional changes in the liquid mixture around the required activity values.

The methodology was then applied to a wide variety of hydrate and solvate solid form equilibria to demonstrate its generality. Hydrates and solvates of theo, nf, cbz and 4ohbzm were studied and the ranges containing the critical activities obtained by CSA-LAG at $25 \pm 1\,°C$ compared to those obtained by slurries at $25\,°C$ (either from literature or performed by us) in Table 2. A variety of active solvents were explored ($h_2o$, dmso, dmf and ace) with ethanol and

ethylene glycol mostly being used as inactive carrier solvents. The thermodynamic boundaries for solvate formation (system-dependent) estimated by CSA-LAG and slurry experiments agree exceptionally well across all systems (Table 2), demonstrating the robustness of the methodology across a broad range of systems. CSA-LAG, however, significantly accelerates the determination of the critical solvent activity boundaries requiring less than a day of experimentation per case study, <1 g of API and <3 ml of solvent mixtures. Additionally, slurry experiments require saturated solutions, which can easily become material intensive depending on the solubility of the compound. CSA-LAG is not only significantly faster than slurry experiments but also requires significantly less starting materials, making it a greener method for generating comprehensive solvate thermodynamic phase diagrams. More importantly, the nucleation of the stable form (solvated or unsolvated) occurs with significant ease in the mill (within seconds to minutes), whereas nucleation from solution can be significantly slower, making it difficult to initiate the conversions and establish equilibrium[16]. CSA-LAG is exceptionally suited for early phase screening, when only small amounts of API are available and when not all solid forms may have been discovered yet.

**Application to sequential solvation (scenario B)**
In this section we apply CSA-LAG to explore the stepwise sequential solvation of 4ohbzm into firstly a HH and then a MH. For stoichiometric solvates, each solvation reaction has an associated critical activity whose determination reveals the regions of existence of the different forms. For the study of the 4ohbzm:$h_2o$ equilibria, we used $h_2o$ as the active solvent and ethylene glycol as the inactive carrier. Figure 5 summarises the solid form outcomes as a function of water activity. A first boundary between 4OHBZM-I and 4OHBZM-HH-I is observed in the activity range of 0.4 – 0.5 and the second boundary between 4OHBZM-HH-I and 4OHBZM-MH-I observed in the activity range of 0.7–0.8. Values determined using competitive slurry

**Table 2 | Solid form equilibrium reactions investigated using CSA-LAG, with corresponding critical solvent activity ranges measured experimentally and compared to those obtained from slurry experiments using the same solvent mixtures**

| Type | Equilibrium reaction | Active solvent | Carrier solvent | $\alpha_s^*$ | |
|---|---|---|---|---|---|
| | | | | **CSA-LAG** ($T = 25 \pm 1\,°C$) | **Slurries[a]** ($T = 25\,°C$) |
| Hydrates | THEO-II + H₂O ⇌ THEO-MH-I | h₂o | ace | 0.50–0.60 | 0.56–0.61[14] |
| | CBZ-III + 2 (H₂O) ⇌ CBZ-DH-I | h₂o | etoh | 0.60–0.70 | 0.60 − 0.63[59] |
| | NF-β + H₂O ⇌ NF-MH-I | h₂o | eg | 0.45–0.55 | 0.45–0.55 |
| Solvates | THEO-II + DMSO ⇌ THEO-DMSO-I | dmso | etoh | 0.20–0.30 | 0.20–0.30 |
| | 4OHBZM-I + Ace ⇌ 4OHBZM-ACET-I | ace | eg | 0.20–0.30 | 0.20–0.30 |
| | CBZ-III + DMSO ⇌ CBZ-DMSO-I | dmso | etoh | 0.20–0.30 | 0.20–0.30 |
| | NF-β + DMF ⇌ NF-DMF-II | dmf | etoh | 0.00–0.10 | 0.00–0.10 |

[a]Competitive slurry critical solvent activities were either taken from the literature (citations given) or determined by us. The NF-MH-I equilibrium data in the table was derived by us with a separate similar value existing in the literature (0.43–0.56[73]).

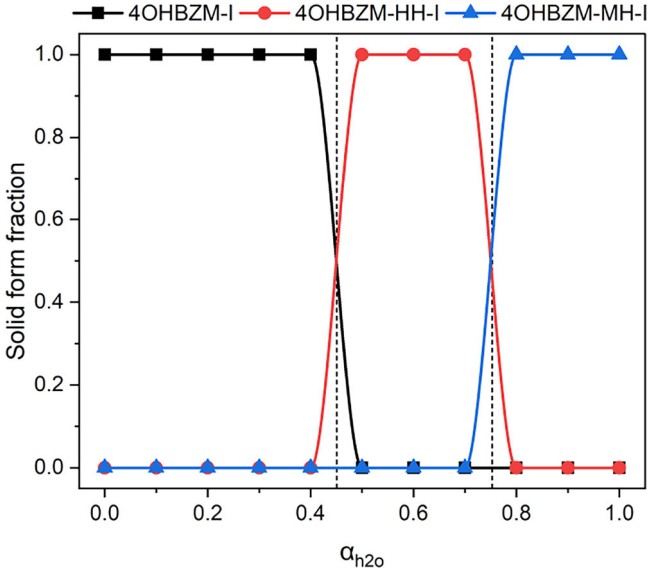

**Fig. 5 | Water activity-dependant phase stability diagram for sequentially solvating compounds.** CSA-LAG phase diagram ($T = 25 \pm 1\,°C$) for the 4ohbzm hydrate system explored using the h₂o: eg liquid mixture and showing the different regions of solid form stabilities as a function of water activity. Critical activities at the boundaries are indicated with dashed lines.

experiments confirmed the results obtained from CSA-LAG (see Supplementary Information, Section 2.2).

### Application to competing solvation (scenario C)

In the third application of CSA-LAG we explore two solvates of the same API with different active solvents competing in the mill. In this scenario both solvents act as 'active' and 'carrier' dynamically, modulating their activities through binary interactions at varying compositions.

We study the conversions between NF-MH-II and NF-DMF-II using a binary solvent mixture containing the two active solvents (h₂o and dmf). Figure 6a shows the conversion from NF-DMF-II to NF-MH-II as the content and activity of water is increased, revealing that $\alpha_{h2o} \approx 0.66$ is required to form the monohydrate. Figure 6b shows the conversion from NF-MH-II to NF-DMF-II as the content and activity of DMF are increased, revealing that $\alpha_{dmf} \approx 0.24$ is required to form the DMF solvate. These values are different from the critical activities observed in the individual solvent equilibria (Section 'Application to single solvation (scenario A)') where only one active solvent is used together

with a carrier solvent, $\alpha_{h2o}^* \approx 0.50$ and $\alpha_{dmf}^* \approx 0.10$ for the hydrate and dmf solvates, respectively. This is as expected since the equilibrium in competing solvention reactions with two active solvents is governed by the ratio of the critical activities (Fig. 1c). For the nf system, the critical activity ratios derived from the individual equilibria for these conversions are: (a) $\alpha_{h2o}^*/\alpha_{dmf}^* \approx 5$ in the dmf solvate to the hydrate conversion and (b) $\alpha_{dmf}^*/\alpha_{h2o}^* \approx 0.2$ in the hydrate to dmf solvate conversion. When plotting the activity ratios for the experimental explorations of solvation competitions in Fig. 6, we observe conversions at activity ratios similar to the anticipated values from the independent equilibria: $\alpha_{h2o}/\alpha_{dmf} > 3$ or the conversion to the hydrate and $\alpha_{dmf}/\alpha_{h2o} > 0.34$ for the conversion to the DMF solvate. The slight discrepancy between the predicted ratio from the critical activities of the individual reactions and that observed experimentally is discussed in Section 'CSA-LAG: methodological considerations'.

### Solid form mapping using a ternary solvent mixture

Solvent mixtures are often used during the manufacturing processes of APIs; therefore, understanding the solvation landscape in multicomponent systems is essential for pharmaceutical product development and fine chemical synthesis. Here, we extend the complexity of our investigations by mapping the solvate landscape of nf using a ternary solvent system. The goal is to map the anhydrous nf as well as the h₂o and dmf solvates across various compositions and activities using an etoh:h₂o:dmf solvent mixture. In this mixture, etoh acts as the 'carrier' solvent allowing for the 'tuning' of the h₂o and dmf activities. Ternary liquid mixtures with defined initial solvent activities were prepared (see Supplementary Information, Section 1.7) and experiments were carried out starting from the anhydrous NF-β form. The results of the series of CSA-LAG outcomes identified using PXRD are illustrated in Fig. 7 (with all data tabulated in the Supplementary Information, Section 2.6), presented as a solvent mol fraction ternary diagram, giving a comprehensive visualisation of the complex solvation landscapes of nf. The map clearly shows the region of stabilities of NF-β (red), NF-DMF-II (green) and NF-MH-II (blue), with well-defined boundaries. The major presence of the green region (NF-DMF-II) is unsurprising since the critical activity for the dmf solvate formation is significantly lower than that of the monohydrate. Similarly, the small red region defining the stability of the anhydrous form (NF-β) is also as expected since both h₂o and dmf are active solvents requiring relatively low activities to form.

Predicting the thermodynamic solid forms in the nf system (NF-β vs NF-MH-II vs NF-DMF-II) in these experiments is inherently complex due to the increased degrees of freedom added using the ternary solvent mixture, necessitating the consideration of the individual solvent activities ($\alpha_{h2o}$, $\alpha_{dmf}$) alongside the solvent activity ratios ($\alpha_{h2o}/\alpha_{dmf}$) to quantify competing solvation driving forces. Compared

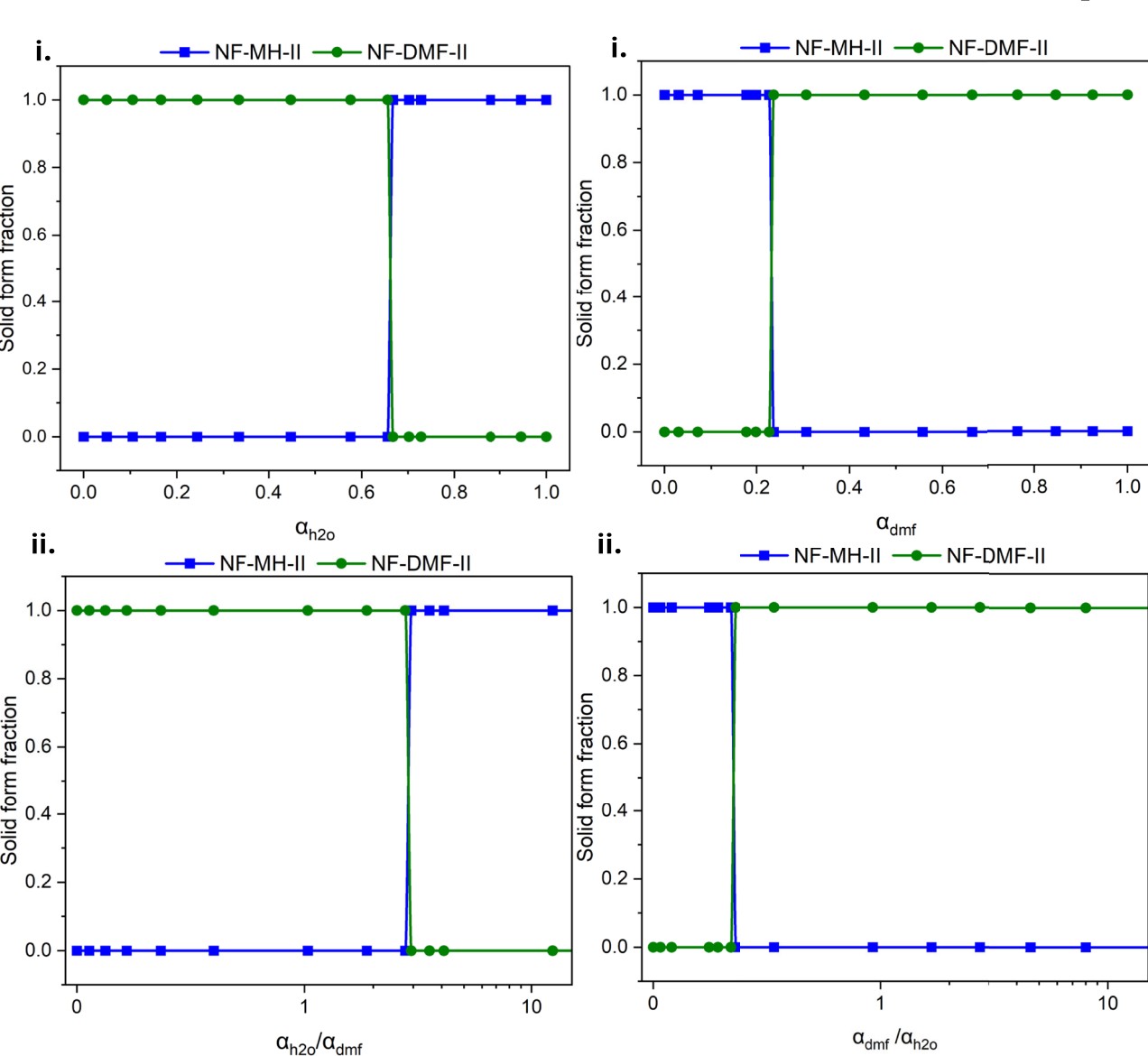

**Fig. 6 | Phase equilibria in competing solvate systems.** CSA-LAG phase diagrams ($T = 25 \pm 1\,°C$) for the nf:dmf and nf:h2o competing systems. **a** Forward equilibrium: NF-DMF-II converts to NF-MH-II with increasing water activity. **b** Reverse equilibrium: NF-MH-II reverts to NF-DMF-II with increasing DMF activity. Subplot (i) shows individual solvent activities and (ii) the activity ratios (plotted on a logarithmic scale).

to Scenarios A, B and C, ternary solvent activity mapping involves intricate balances between solvent activities and form stabilities. Despite challenges in correlating initial solvent properties with steady-state CSA-LAG milling outcomes, activity-based rules remain predictive, offering insights into the complex multicomponent solvation landscapes.

### CSA-LAG: methodological considerations

We have shown that CSA-LAG is a fast reliable and robust method to determine boundaries of solvate formation in single, sequential and competing solvation reactions. The agreement with the boundaries identified with the more "traditional" slurry experiments for all systems studied here is excellent as shown from the linear correlation in Fig. 8 with $R^2$ factor of 0.975.

The excellent agreement between slurries and CSA-LAG is unsurprising since both experimental approaches drive the studied systems to full thermodynamic equilibrium. We note however that

some literature exists reporting disagreements of solid form outcomes between solution crystallisation methods and LAG for a variety of systems[47,53]. In such cases kinetic effects are likely at play in solution crystallisation—especially if equilibration via slurrying is not achieved. In our experience with solvates we have sometimes obtained meta-stable solvates from solution crystallisation which cannot be obtained by CSA-LAG and we have also produced stable solvates from CSA-LAG that cannot be produced from solution (even slurries). The fact that thermodynamic equilibrium is rapidly and consistently achieved via LAG naturally makes CSA-LAG a more robust methodology for phase diagram generation than solution methods.

Since the method uses milling, equilibrium is achieved between solids with very small particle sizes—in the range of nanometres[40]. Because of this some small experimental discrepancy may be observed when comparing our CSA-LAG method to other suspension methods (solution-mediated slurries) that lead to much larger crystals, simply due to different length scales they operate in. Analogously, solution

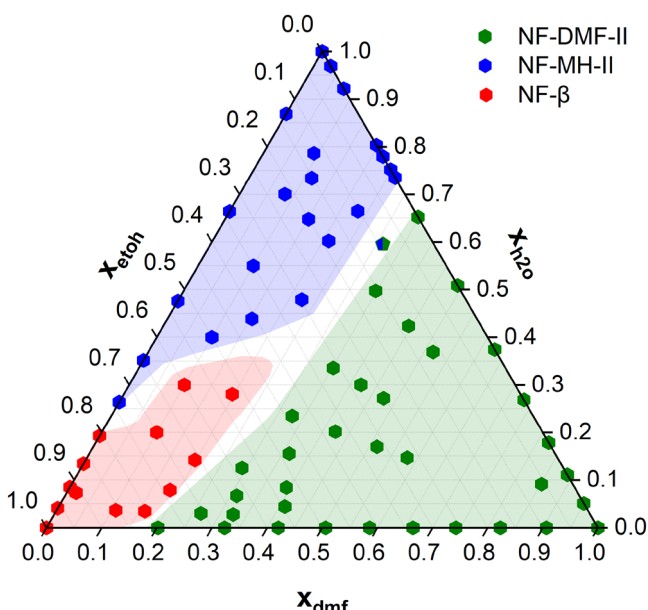

**Fig. 7 | Ternary solvent phase diagram for nitrofurantoin determined by CSA-LAG.** CSA-LAG outcomes for nf in a ternary solvent system (h2o, dmf and etoh). Blue marks NF-MH-II (monohydrate), green NF-DMF-II (DMF solvate) and red NF-β (anhydrous). Points indicate experimentally identified forms; colour-mapped regions denote thermodynamically stable phases. Plot shown in mole fraction (not activity), as $\alpha_{dmf} + \alpha_{h2o} + \alpha_{etoh} \neq 1.0$.

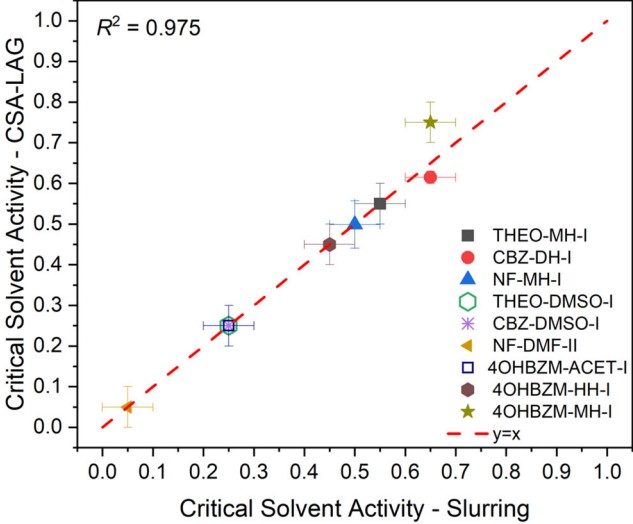

**Fig. 8 | Validation of CSA-LAG against traditional slurry methods.** Comparison of the critical solvent activities obtained from CSA-LAG (y axis) versus slurrying (x axis). A perfect linear fit y = x with an $R$ squared value of 0.975 indicates a very strong agreement for the thermodynamic boundaries obtained from the two different methods. Horizontal and vertical error bars represent the resolution of phase boundary determination, showing the range of controlled water activities tested above and below the transition point.

methods will result in larger crystals with specific shapes which may lead to significant preferred orientation upon PXRD analyses, whilst CSA-LAG produces much smaller crystals which reduces the scale of preferred orientation allowing for an easier PXRD identification.

Throughout all CSA-LAG experiments and PXRD analyses carried out in this work (147 diffraction experiments), we have observed no obvious evidence of amorphisation. Whilst amorphisation is possible by milling, the conditions of milling are critical. For example, neat milling can lead to amorphisation in some systems, whilst milling in the presence of solvent promotes the formation and growth of crystals in the mill[26,32,40,54]. Since our method uses milling in the presence of solvent, we have seen no obvious evidence of amorphisation from the diffraction patterns. We note that the raw PXRD patterns contain characteristic 'halo bumps' due to the use of the Kapton tape to cover the sample (Supplementary Information, Section 1.8.1) and not amorphization and that upon applying a Kapton background scattering corrections, the resulting patterns have very well-defined diffraction peaks (as shown in Fig. 4 and the Supplementary Information, Figs. S6 and S7). Whilst thermal analysis can help quantify amorphisation it cannot easily be used for our milled solids since they are wet, which would lead to complex DSC thermographs containing several events including solvent evaporation and phase transitions of difficult resolution and interpretation. Further to this for amorphisation to occur in the mill several conditions need to be satisfied: (a) the solid forms under study must have a $T_g$ above the milling temperature, (b) high intensity milling is required and (c) low humidity must be ensured since water (even in small amounts) acts as a strong plasticiser. Given that our milling is performed at room temperature, is of relative low intensity and duration and is carried out in the presence of solvents (with their consequential plasticising effects[55] and enabling crystal growth in the mill[26]), amorphisation under our experimental conditions is highly unlikely. Our observations are consistent with prior literature on the milling of hydrates[56] which states of the 'impossibility to amorphise most hydrates by milling due to the plasticising effect of the structural water molecules which generally depresses the $T_g$ of the

corresponding amorphous forms below the milling temperature'— with similar expectations for solvates[55].

Since the stability of solvates is intricately dependent on environmental conditions, the experimental protocols need to be carefully executed to avoid unexpected changes in the liquid compositions leading to changes in activities and unexplained conversions. This is particularly important when the active solvent is volatile. In our early works on CSA-LAG we observed ex-situ desolvation of the forms obtained by milling prior to analysis or during PXRD acquisition. Carefully covering the milled powders retrieved from the milling jar with Kapton (Fig. 2) prevents these conversions. This simple yet effective strategy enables CSA-LAG to be performed even in systems where rapid solvation or desolvation kinetics are involved, facilitating accurate and controlled analysis. This procedure then requires the subtraction of the background Kapton scattering contributions from the PXRD analyses but ensures solid form preservation.

The activity in the prepared mixed solvents can change significantly with very small changes in the solvent composition, regardless of the solvation exploration method used. Since we work at very small scales (μL), it is very important to apply special care when preparing, handling and storing those mixtures. In that regard, HPLC grade and higher boiling point solvents (ethylene glycol and propylene glycol) are preferred and recommended to minimise effects associated with component evaporation. Furthermore, solvents and solvent mixtures should be stored under controlled conditions to prevent water uptake or solvent loss due to exposure to the environment. To limit the unintentional water content from environmental uptake, if solvents cannot be purchased dry, solvents should be pre-dried using 3 Å molecular sieves[57].

Small shifts in the activities of solvent mixtures due to inherent experimental uncertainties can lead to enormous shifts in the activity ratios. For example, we found we needed to be especially careful with experimentations carried out for the competing solvate work in scenario C. The predicted critical activity ratios can differ significantly from those observed experimentally simply because a 5 mol% experimental uncertainty can lead to an order of magnitude change in the activity ratio. To reduce those uncertainties, one can scale up the

amount of solvent used in the CSA-LAG. For scenario C, studying competing solvates work, we found that working with $5\,\mu L$ per mg of API solids (rather than $2\,\mu L$ per mg) helped reduce these uncertainties and led to observed activity ratios for solvate conversions more similar to those predicted from the critical activity values of the independent equilibria.

In addition to the compositional effects on solvent activity in CSA-LAG experiments, temperature, also plays a critical role governing the relative location of the thermodynamic boundaries of hydrates and solvates. Whilst our Retch MM400 mill does not have the capability to monitor or control the temperature, several precautions were taken to maintain a consistent temperature range across CSA-LAG experiments. To this effect the milling jars along with their content were always precooled to 20 °C prior to milling; this led to final jar temperatures (post milling) in the range of 24–26 °C (Supplementary Information, Section 2.4). The jars were removed from the MM400 mixer mill hands immediately after milling, as the residual heat dissipating from the motor can lead to temperature transfer post milling. Subsequently the mill was allowed to cool down with its lid left open for approximately 60–90 mins between milling experiments, which we found to yield consistent final milling temperatures.

Another advantage of CSA-LAG and mechanochemistry in general is concerning the excellent mixing and homogenisation achieved during milling[34,35]. This contrasts with conventional solution slurries, where the mixing efficiency is more variable. In fact, a recent study by Cherepanova et al. showed that reaction kinetics and reproducibility can be dependent on the position of the stirrer bar relative to the plate; meaning that even such a routine parameter can influence the potential equilibration time for the case of solvates[58]. By comparison, the mechanochemical route of CSA-LAG, though milling actively contributes towards homogeneity[35], offering a more controlled pathway to probe solvation equilibria.

Over the course of the CSA-LAG experiments carried out in our study mixtures of phases were only observed under a limited number of milling conditions (Supplementary Information, Section 2). There are three important considerations that account for these mixtures. First, if one of the liquid-mixtures used for the LAG has a solvent activity very close to the critical value, it is possible that the solids at equilibrium contain both solvated and unsolvated phases, since both are equally stable at the critical activity. Second, we have used a liquid to solids ratio of 2 μL/mg which means that the liquid is in overall molar excess relative to the solids by about a factor of 16–30 (depending on the system). However as solvent mixtures are used and composition of the active solvent and its activity is progressively increased, there exists a small window of activities where the number of mols of active solvent fall slightly below the stoichiometric requirement for full solvation, which can lead to mixed phases. This is rare since the windows of activities where this may occur more commonly will lie between 0 and 0.2 in most cases. In the case of competing solvation (scenario C) this was overcome by using a very high liquid to solids ratio of 5 μL/mg bypassing stoichiometric limitations. Third, for instances where the API has a higher than usual solubility in both the active and the carrier solvent, the liquid to solid ratio needs to be reduced in order to avoid full dissolution. These conditions will lead to wider windows where the active solvent to API stoichiometries will lie below the stoichiometric reaction needs. We encountered this issue only for one of the systems: 4ohbzm in CSA-LAG with ace:eg. The unusually higher solubility of both the anhydrous and acetone solvate in the liquid mixture meant that CSA-LAG could only be effectively performed by lowering the liquid to solids ratio to 1 μL/mg which in turn led to a window of activities for which mixtures of the anhydrous form and the acetone solvate were observed. The critical activity always lies between the last activity that gives the pure anhydrous form and the first activity that leads to solvation (independent on whether the outcome is the pure solvate or a mixture).

Furthermore, the critical water activity for a an/hydrate pair is mixture irrelevant as evidenced by the boundary comparison measured by CSA-LAG and slurrying across distinct carrier solvents (Supplementary Information, Tables S17 and 18).

This study implements controlled activities in liquid-assisted grinding, representing a methodological innovation and a key strength of our approach. Using activities rather than concentrations is essential for capturing the non-ideality of solvent mixtures, ensuring transferability across solvent mixtures containing the same active solvent but a different carrier solvents (Supplementary Information, Section 2.7), enabling meaningful comparisons between methodologies and allowing prediction of solid form transformations under varying environmental conditions. For example, the critical water activity for the NF-MH-I transition, measured using CSA-LAG in a h2o:eg solvent mixture, is approximately 0.5—corresponding to a water mole fraction of 0.52 in that system (Supplementary Information, Section 2.7). This activity threshold defines the thermodynamic limit for hydrate formation and remains consistent across solvent mixtures and experimental methods, thereby enabling robust prediction of behaviour. If nf is developed in its anhydrous form using a h2o:ace mixture, maintaining water activity below 0.5 ensures the stability of the anhydrate; however, relying on mole fraction alone (e.g., 0.52) would be misleading, as this corresponds to a water activity of 0.76 in h2o:ace, promoting hydrate formation. Furthermore, the critical activity determined via CSA-LAG can be extended to predict behaviour under liquid-vapour equilibrium conditions, such as relative humidity for water, whereas concentrations lack this transferability.

Finally, we note that our procedure always starts with the solids of pure APIs and not their solvates. This is important since we are working at small scales and the solvent released from a solvate under milling forces will impact the activity of the initial liquid. Hence, for accuracy and simplicity, the mapping of the phase diagrams is done by exploring solvation reactions only.

Regarding scope and future directions, CSA-LAG maps solvation boundaries by varying bulk solvent activities obtained from VLE data, while the API remains largely a solid. This pragmatic assumption for non-electrolytes has been standardised across the crystallisation field over the last three decades for hydrate/solvate equilibria[1,15,16,18,50,51,59] and agrees with independent slurry equilibria across our set ($R^2 = 0.975$, Fig. 8), indicating that the endpoints reached by LAG are thermodynamically meaningful. We recognise that LAG milling can lead to high interfacial areas and localised transient supersaturation, where system-dependent short-lived deviations from the bulk activity can be expected. Our analysis, however, relies on the equilibrated solids post milling rather than on the transient pathway or intermediates.

Whilst we acknowledge that high dissolved fractions can effectively perturb the bulk activity in comparison with the predicted NRTL activities, this represents an opportunity for methodological refinement rather than a limitation. For most pharmaceutical systems, low-moderate solubilities ensure negligible perturbation of solvent activities, well within typical VLE model uncertainties. In exceptional cases of higher solubilities (eg. 4OHBZM-acetone), we employ practical mitigation strategies: reduction of liquid:solid ratio, or selection of passive solvents that suppress dissolution. Where high solubility is unavoidable, NRTL derived activities still remain informative predictors of phase behaviour, with appropriate wider uncertainty bounds.

Moving forward, several opportunities exist to further enhance the methodology and the general understanding of hydrates. First, direct headspace activity measurements would provide explicit validation and understanding to quantify and potentially model solubility induced perturbations. Second, temperature-controlled milling using jacketed grinding jars would enable systematic mapping of critical activities as a function of temperature, decoupling thermal from compositional effects and expanding CSA-LAG use for full thermodynamic phase diagram construction. Third, time resolved in situ

synchrotron diffraction experiments under CSA-LAG will further demonstrate the efficiency of our method and shed light on the transformation pathways. These future studies, which are ongoing, would build upon CSA-LAG's demonstrated ability to rapidly and reproducibly map solvate landscapes, transforming hydrate screening from a month-long trial and error to a single day thermodynamically guided approach.

To conclude, we have introduced controlled solvent activity liquid-assisted grinding (CSA-LAG) as a scalable, robust, efficient and adaptive mechanochemistry method for mapping the thermodynamic landscapes of stoichiometric solvates and hydrates. Enabling the control of solvent activities by using liquid mixtures of precise compositions, CSA-LAG paves the way for the construction of complex phase diagrams across a range of solvation scenarios with exceptional accuracy. We have shown the generality of the technique by constructing phase diagrams for four pharmaceutical compounds forming hydrates and solvates with a variety of solvents and at varied stoichiometries. We have also shown that CSA-LAG can be used to study competitive solvation in binary as well as ternary liquid mixtures.

Our work establishes CSA-LAG as a robust method for probing crystallisation thermodynamics via mechanochemistry within minutes, setting a benchmark for the rapid, resource-efficient mapping of solvate landscapes. We anticipate that CSA-LAG will significantly influence crystallisation workflows in industry and academia, enabling the systematic exploration and potential exploitation of previously overlooked solvate forms.

## Methods

### Materials
Nitrofurantoin anhydrous (purity > 99%), theophylline anhydrous (purity > 99%), carbamazepine anhydrous (purity > 99%), and 4-hydroxybenzamide anhydrous (purity > 98%) were all purchased from Sigma-Aldrich. Commercial solid forms were phase identified by PXRD and were subsequently used without any further processing. Dry organic solvents including acetone (HPLC), N,N-dimethylformamide (HPLC), ethanol (HPLC) and ethylene glycol (HPLC) were purchased from Sigma and were used without further purification. Demineralised ultrapure water was purchased from Fisher Scientific.

### Solvent activity calculation
For the calculation of the desired solvent activities for binary and ternary solvent mixtures targeting the formation of hydrates, solvates, or anhydrous phases, the non-random-two-liquid model (NRTL) was used as implemented in ASPEN Plus v14.1. The NRTL model was selected as it is applicable for ambient environmental conditions, complete miscibility of components and multi-component interactions[60,61]. The model parameters are defined from fitted experimental data of steady-state vapour liquid equilibria (VLE), available in the NIST, ASPEN, DECHEMA and IG databases. Alternatively other calculation approaches include the Wilson Theory (UNIversal QUAsi Chemical model) UNIQUAC and (UNIversal Function Activity Coefficient Model) UNIFAC and COSMO-RS all of which are commonly used by engineers to model stream properties in the chemical and pharmaceutical industries[60,62]. Here the calculated solvent activities represent the bulk solvent mixtures. Since the API remains largely in the solid-state during CSA-LAG solvent-solvent interactions dominate and the NRTL derived activities provide a valid approximation, as also applied in slurry-based studies[1,14,16,49,51,59] and industrial crystallisation practice. Any local perturbations induced by solute-solvent interactions would only slightly shift the observed thresholds, which are implicitly accounted for in the experimental phase boundaries reported.

### Preparation of solvent mixtures
Solvent selection for the mixtures must allow complete miscibility of all components in all ratios allowing for a uniform activity throughout the liquid and vapour phases. Solvent selection is based on the crystallisation outcome of the API from the desired solute, whether that would be anhydrous, hydrate or solvate. All liquid volume handling was performed using Eppendorf Research Plus micropipettes with single use tips for all decanting (Eppendorf, Hamburg, Germany). The resulting mixtures were vortex-mixed for one minute after preparation and every time prior to milling to ensure homogeneous mixing and thus activity uniformity throughout the solution. Mixtures were typically prepared on a 10 ml scale to account for evaporative losses, stored in screw top vials with polyvinyl faced liners and tightly wrapped with parafilm along the cap and glass seal to minimise solvent losses. Weight monitoring of the vial is suggested after long-term storage.

### CSA-LAG
CSA-LAG experiments were conducted using a Retsch MM400 Mixer Mill with screw top 10 ml zirconium oxide milling jars and a single 10 mm ball of zirconium oxide, fitted with a Teflon gasket to ensure an environmental seal. Approximately 100 mg of sample and 200 μL of (CSA) solutions were used. After preparation of the jars they were precooled in the fridge until their temperature reached 20 °C (as measured with an infra-red thermometer). At that point, the jars were loaded onto the MM400 and the samples were milled at a frequency of 30 Hz for 30 min. Immediately after milling, the jars were removed and the temperature monitored on the jar surface using an infrared thermometer (ETEKCIT Lasergrip 1080). In all experiments, the recorded temperature ranged from 24 to 26 °C. Subsequently the mill was allowed to cool down with its lid left open for approximately 60–90 mins between milling experiments, which we found to yield consistent final milling temperatures.

### Controlled solvent activity solvent mediated phase transformation (slurries)
For the determination of the critical water/solvent activities at a constant temperature the method of controlled solvent activity slurry bridging was implemented. The prepared solvent mixtures of desired activities were saturated with the anhydrous form of the API of interest. Next the solutions were supersaturated by the addition a 50:50 mixture of the two competing forms (i.e. solvate and anhydrous), to allow the bridging of the two forms without the need for nucleation (a kinetically influenced process). The resulting suspensions were sealed with PTFE screw top lids and tightly wrapped with several parafilm layers to limit solvent permeation. Using magnetic stirrer bars the solutions were agitated at a constant rate and temperature (25 °C) using the Polar Bear Plus Crystal (Cambridge Reactor Design, Cambridge, United Kingdom). After two weeks, the weight of the vials was measured to confirm that no significant solvent loss had occurred and the resulting precipitated crystals were analysed via PXRD.

### Powder X-ray diffraction (PXRD)
PXRD patterns of the milled and slurried samples were acquired on a Brucker AXS D8 Advance diffractometer fitted with a LYNXEYE PSD Detector, in Bragg-Brentano geometry using Cu K$a_{1,2}$ radiation and a Ni filter. A constant 12 × 6 mm sample area was illuminated during measurements. Data were collected on a 2θ range of 2–50° with a time step of 0.5 and a 2θ increment of 0.02° per step. Samples were analysed on Si low background wafer mounts with a sample depth of 0.1 mm (-20 mg in total). To minimise solvent loss samples were covered with 0.01 mm Kapton (polyimide) film. All samples were prepared and analysed immediately after the opening of the milling jars, to limit changes to the activity at the end of the milling process. Since the use of Kapton introduces broad scattering contributions together with those of the Si zero-background puck, separate background patterns were collected and subtracted from the raw data. This procedure removes artefacts and yields the intrinsic diffraction patterns of the

crystalline samples. A further description of the PXRD sample preparation can be found in the Supplementary Information*.

## Rietveld refinement

Quantitative analysis was performed using the Rietveld method using TOPAS ACADEMIC version 8[63–66]. Structural models for BAPLOT01 and THEOPH05 were taken from the CCDC and fractional coordinates fixed. A single overall isotropic temperature factor was refined for all atomic sites. The complex background due to the sample holder was modelled using 16 parameters. One parameter was used to scale a separately-measured scan of the empty holder to the data; an additional 9 parameters were used to describe Voigtian functions to fit broad Kapton arising background 'humps'; and an additional 6 parameters described a smoothly-varying Chebychev polynomial. Preferred orientation was corrected using a March Dollase 1 parameter preferred orientation model. Rietveld-extracted weight percentages were 29.5(5)% and 70.5(5)% for BAPLOT01 and THEOPH05 respectively.

Structural models for LABJON02 (room temperature redetermination of LABLON) and 2443102 (NF-DMF-II) were taken from the CCDC and fractional coordinates fixed. A single overall isotropic temperature factor was refined for all atomic sites. Peak shapes were described using a Thomas-Cox-Hastings pseudo-Voigt function with axial asymmetry and variable divergence intensity. The background was modelled with a 6 smoothly-varying Chebychev polynomial and an additional $1/x$ term to capture low-angle scattering. Preferred orientation was corrected using 4th order spherical harmonics for each phase. Crystallite size and microstrain contributions were also refined. Rietveld-extracted weight percentages were 28.4(5)% and 71.5(5)% for 2443102 (NF-DMF-II) and LABJON02 respectively.

## Single crystal X-ray diffraction (SCXRD)

SCXRD was conducted for the structural elucidation of 4-hydroxybenzamide acetone solvate (4OHBZM-ACET-I) and nitrofurantoin dimethylformamide solvate (NF-DMF-II). Information regarding the crystallisation of single crystals of these forms is included in the Supplementary Information*.

The X-ray single crystal data for 4OHBZM-ACET-I were collected at a temperature of 120.0(2) K using MoKα radiation ($\lambda = 0.71073$ Å) on a Bruker D8 Venture with a Photon III MM C14 CPAD detector, IμS-III-microsource, focusing mirrors diffractometer equipped with a Cryostream (Oxford Cryosystems 700) open-flow nitrogen cryostat. The data for NF-DMF-II were collected at 100.0(2) K at the I-19 beamline (Dectris Pilatus 2M pixel-array photon-counting detector, undulator, graphite monochromator, $\lambda = 0.68890$ Å) at the Diamond Light Source, Oxfordshire and processed using Xia2/DIALS[63,65,67–69].

The structures were solved using Olex2[70] with the ShelXT[71] structure solution program using intrinsic phasing and refined with the ShelXL[72] refinement package using Least Squares minimisation on $F^2$. All non-hydrogen atoms were refined with anisotropic displacement parameters. Hydrogen atoms were located in the difference map and refined isotropically using a riding model unless otherwise specified. Crystal data and refinement parameters are provided in the Supplementary Information. Crystallographic data for 4OHBZM-ACET-I and NF-DMF-II have been deposited with the Cambridge Crystallographic Data Centre with deposition numbers CCDC-2443101 and CCDC-2443102, respectively.

## Data availability

The authors declare that the data for materials, solvents, thermodynamic properties and results supporting the findings of this study are available within the article and its Supplementary Information. The resolved crystal structures have been deposited with the Cambridge Crystallographic Data Centre (CCDC deposition numbers 2443101 and 2443102). Data are available from the corresponding author upon request.

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

## Acknowledgements

F.T. and A.J.C.C. thank Pfizer Inc. and the UK Engineering and Physical Sciences Research Council for supporting and funding this work through a CASE award (grant number EP/W524426/1 F.T.). F.T. and P.B. thank Ioannis Asproudis for the initial experimental estimation of NF-β water activity performed at Pfizer. We thank Dr. Wesley Clack for all the valuable discussions. F.T. and T.J.B. acknowledge and thank the beamline scientists Dr. Dave Allan and Dr. Sarah Barnett from Diamond Light Source for access and use of beamline I-19 under BAG proposal (CY30280 T.J.B.). F.T. thanks Adam Michalchuk and Giulio Lampronti for the valuable discussions. F.T. and A.J.C.C. thank Gary Oswald and the Durham X-Ray powder diffraction service for the extensive access to PXRD facilities. We also thank Prof Sharon Cooper for the valuable discussions.

## Author contributions

F.T. and A.J.C.C. designed the work and developed the ideas as well as the experimental procedure and led the writing of the manuscript. F.T. performed all of the experimentation of the CSA-LAG work. T.J.B. and J.S.O.E. contributed to the extensive characterisation of the systems. P.B. and N.F. contributed to conceptualisation, expertise and insightful project directions. All authors contributed to the writing of the manuscript.

## Competing interests

The authors declare no competing interests.
