## [Transparent Peer Review File · Nature Communications]

Phase Diagrams of Pharmaceutical Solvates from Mechanochemistry

Corresponding Author: Professor Aurora Cruz-Cabeza

Version 0:

Reviewer comments:

Reviewer #1

(Remarks to the Author)

This is a very interesting article that opens the door to numerous similar studies and, hopefully, encourages greater use of mechanochemistry as a screening tool, which is more sustainable than slurry methods. I greatly appreciate the article's ability to synthesize complex information—a task that is certainly not easy—both in textual form and through the use of equations and graphical representations.

However, in my opinion, the manuscript would benefit from providing clearer guidance to the reader regarding the rationale for the choice of solvents and the four APIs, as well as for the selection of their solid forms. It also lacks an early explanation of the role of the solvating solvent and the inactive carrier. The role of the solvent, including as a solvating solvent in LAG, remains insufficiently understood in the scientific literature, which makes it even more important to explain it clearly here—particularly if the authors are confident in their interpretation.

In summary, it would be valuable to include a concise introductory section that sets the stage before presenting the core of the work. This would help guide the reader more smoothly through the subsequent sections. The discussion does not adequately clarify the theoretical basis for the rationalization of the three scenarios in the equations. As a result, the text delves straight into the subject matter without providing an adequate introductory section to guide the reader. As it stands, the article remains rather dense and could be challenging to follow without this context. Finally, while the work is interesting, the overall approach is not entirely novel, which further highlights the importance of emphasizing what is truly new in this study

Suggestions for Improvement

- **Clarify Experimental Rationale:** The article would benefit from more guidance regarding the selection of solvents, APIs, and their solid forms. An early explanation of the role of the solvating solvent and the inactive carrier is missing, and its absence makes the manuscript harder to follow. A brief introductory section outlining the rationale behind these choices would greatly improve readability and accessibility.
- **Process Conditions and Temperature Control:** The process conditions are not clearly specified, and the claim that the temperature remains at 25 °C during milling (30 Hz for 30–45 minutes) seems implausible. Mechanochemical reactions are highly sensitive to temperature and other process variables. More precise measurement or control of these parameters is essential. The authors should provide clarification and, ideally, include temperature monitoring data.
- **Consistency Between Main Text and Supplementary Information:** The Supplementary Information (particularly section 1.3) contains discrepancies in milling time, powder quantity, and liquid volume compared to the main text. These differences should be addressed, as they compromise reproducibility. Moreover, the gravimetric solubility measurements lack key details—most notably, the temperature at which they were conducted. This omission is critical, given the temperature dependence of solubility.
- **Graphical Abstract and Data Presentation:** The graphical abstract does not effectively convey the study's focus. I recommend revising it to better reflect the core findings, maybe starting from the cscheme in Figure 2. Additionally, a summary table listing the APIs, solvents, and solvent mixtures used—including composition ranges for binary and ternary systems—would be extremely helpful for readers navigating the experimental design.
- **Structural Clarity:** The Supplementary Information includes a well-organized index that could be partially integrated into the main text to improve the logical flow. This would help guide readers through the dense content more effectively.

OTHER SUGGESTIONS:

- A more comprehensive discussion of the relevant literature would have been also appreciated, rather than treating it only briefly. For example: Cruz-Cabeza, A. J.; Wright, S. E.; Bacchi, A. On the Entropy Cost of Making Solvates. *Chem. Commun.* 2020, 56 (38), 5127–5130, DOI: 10.1039/D0CC01050B; Conformational Change in Molecular Crystals: Impact of Solvate Formation and Importance of Conformational Free Energies DOI: 10.1021/acs.cgd.1c00833; Lombard, J.; Laker, H.; Prins,

- A table summarizing all the boundaries in both mechanochemical LAG and slurry experiments is strongly recommended.
- The observation of results identical to those obtained by slurry is interesting; however, based on my experience, this is not very common. In fact, certain solvates formed mechanochemically do not typically form via slurry methods. I would encourage the authors to emphasize this distinction and discuss the implications of this finding in greater detail.
- This table is not clear: Table S7 — Ternary molar composition of DMF, water, and ethanol used to prepare mixtures of known initial solvent activity. Please clarify the meaning of all acronyms used, either directly in the table or in a footnote, to ensure that the information is readily understandable to the reader.
- Rietveld refinement is not mentioned in the Materials and Methods section, but only in the Supplementary Information and in the Results. It should be described in sufficient detail in the Materials and Methods in the main paper.
- Is it correct to understand that no mixtures of solid forms are obtained at any given solvent activity? From Tables S13...it seems that only single solid forms are reported. Clarification on whether solid form mixtures occur—or why they do not—would be valuable for interpreting the results accurately. Although this is reported for scenario C, it is quite remarkable that in the other cases no sample appears to deviate from ideality. Given that there is often a discrepancy between theory and practice, this point would deserve further discussion.
- The table listing Crystal Form 4OHBZM-ACET-I and NF-DMF-II, along with their crystallographic data and CCDC deposition numbers (2443101 and 2443102), appears to report structural details of newly resolved crystals rather than an overview of thermodynamic boundaries as indicated by the title of Table S28 Table S 28: Overview of the thermodynamic boundaries of the hydrates and solvates studies with CSA-LAG and slurring. Please clarify the intended content and labeling of this table to avoid confusion.
- ABSTRACT: Scientific acronyms are quite fashionable these days — it seems every researcher needs to be remembered by one: “I was the one who introduced this technique” CSA-LAG... well, I'm not entirely convinced. Perhaps an acronym with a smoother flow could be found. Of course, this is a matter of personal preference.

AI suggested to me:

GRASP – Grinding under Regulated Activity of Solvent Phase

SAM – Solvent Activity Milling

SAGE – Solvent Activity Guided Experiment

SAGE-M – Solvent Activity Guided Experiment – Milling

SARM – Solvent Activity Regulated Milling

- LINE 32 It is not clear to me — are there two graphical abstracts? In my opinion, the concept of CSA is not clearly highlighted, nor is the motion of the jars effectively represented. A modification of Figure 1 that continues the storyline beyond the PXRD analysis — showing the construction of the phase diagram and the application of the three-scenario equations — would be an improvement.
- LINE 79 I would suggest including this article as well: <https://dx.doi.org/10.3390/cryst14040374>. Although it was published in a journal that is not particularly prestigious, given the limited literature available on this topic, I believe its inclusion is appropriate.
- LINE 94: The meaning of m, n, ... w ... in relation to the stoichiometry is not entirely clear. Given that solvent mixtures are being added, it is unclear how these variables are defined and what their specific stoichiometric significance is in this context.

Some information can be taken from the ESI and included in the main. In my opinion, what remains particularly unclear is the stoichiometric ratio between the solid and the solvent, whether the solvent is used alone or in mixtures. This lack of clarity is further compounded by the inconsistent reporting of the solid-to-liquid ratio throughout the manuscript. The experimental section in the main text (line 449 and following), the ESI (1.2. milling pag. 3), and the results discussion each refer to different ratios for the LAG conditions, making it difficult to assess the reproducibility and interpret the mechanistic implications of the solvate formation. A clear and consistent definition of the stoichiometry used is essential for evaluating the validity of the conclusions.

- LINE 98 Does the equation really work for all solvates? Wouldn't it be more accurate to state that it “is expected” to work for all types of stoichiometric solvates? Does it really work also for mixed (hetero) stoichiometric solvates containing two solvents? If not, which equation would apply in that case?
- LINE 175: this sentence must be written near Table 1. For a reader with expertise in pharmaceutical solid forms, the rationale for selecting these particular solid forms is not entirely clear, especially considering that multiple forms are known for each of these APIs
- LINE 181: please note that this statement appears to be inconsistent with the choice of acetone.
- LINE 224 In my opinion, this sentence would be more appropriate in the Introduction section.
- LINE 256 In my opinion, this sentence would be more appropriate in the Introduction section.
- LINE 308 in water formula the subscript 2 is not visible
- LINE 334 Before linear regression, a table summarizing all the boundaries in both mechanochemical LAG and slurry experiments is strongly recommended. Please note that the ESI appears to contain an incorrect boundaries table, as it does not include the boundaries but the crystal parameters of the new solvates. So, in fact, the boundaries are not reported anywhere. It is essential that the correct data be provided to ensure transparency and allow for meaningful interpretation and replication of the results.
- FIGURE 8 : "solvant" instead of "solvent"
- LINE 357: Slurry are suspensions not solution.
- LINE 365: It is widely recognized that liquid-assisted grinding (LAG) typically does not yield amorphous forms. Therefore, this aspect does not seem surprising. However, it is somewhat surprising that no mixed solvates solvates have been observed in this study. Clarification or discussion on this point would enhance the manuscript.
- LINE 442: please explain the choice of the 6 solvents

- LINE 447 Please specify the container in which prepared solvent mixtures were stored and the type of lid used.
- LINE 450 Please specify the number of balls used and the type of material. From the graphical abstract, it appears that a single ball was used and that it is made of a different material than the jar.
- LINE 451: in the results section you stated 2 microliter for each microgram. it is contradictory with this sentence.
- LINE 453: here you mentioned 2-30 min milling time; in the Results section only 30 min is mentioned. Then in ESI 30-60 min. It is unfortunate to find such divergent experimental conditions reported for mechanochemical reactions, which are known to be highly sensitive to process variables. This is particularly concerning in a study that claims the temperature remains constant at 25 °C throughout. Without consistent and well-defined parameters, the reliability of the mechanistic conclusions is compromised.
- LINE 457 Please specify how the temperature was measured — during the milling process or after opening the jar at the end. While it is true that short milling times, even at 30 Hz, may result in temperatures around 25 °C, in my personal experience and from literature data, milling for 45 minutes at 30 Hz significantly exceeds 25 °C and can reach approximately 35-40 °C.
- Line 457 Please specify the temperature. A similar lack of attention to experimental detail is evident in the gravimetric solubility measurements, where crucial parameters such as temperature are not even mentioned (see ESI section 1.3). Given the temperature dependence of solubility, this omission raises concerns about the reproducibility and accuracy of the reported values.

Reviewer #2

(Remarks to the Author)

The manuscript "Shaken, not stirred! Phase diagrams of Pharmaceutical Solvates", by Cruz-Cabeza et. al., describes a new method based on mechanochemistry for screening the formation of solvates and hydrates of pharmaceutical compounds using the solvent activity of solvent mixtures. Although the proposed method demonstrates advantages over slurry methods for screening new solvates and hydrates, mechanochemical methods, particularly ball mill grinding, are already well-established. These methods have been employed to screen the formation of both crystalline (e.g., polymorphs, co-crystals/salts) and amorphous forms of pharmaceutical compounds by varying conditions such as solvent type, ball mill type, milling time, frequency, and the number and size of milling balls.

The novelty of the present work lies on using the solvent activity of solvent mixtures to define the experimental conditions under which a compound will form a hydrate or solvate. While the method can be extended to other pharmaceutical compounds, a screening process remains necessary depending on the specific system studied. Moreover, when scaling up the process, particularly for industrial applications using e.g., industrial-scale mills, the experimental conditions optimized for the studied model systems (theophylline, 4-hydroxybenzamide, carbamazepine, and nitrofurantoin) are likely to change. Even if a hydrate form is identified under specific laboratory-scale conditions, it may not be reproducible upon scale-up. Additionally, any adjustments to the scale-up process may lead to the formation of other solid-state forms. Did the authors consider varying conditions such as the amount of solvent (fixed here at 200 μ L), milling time, and type of ball mill (e.g., using a planetary mill) to investigate how these factors influence the obtained results?

The work demonstrates potential for a more sustainable approach to screening pharmaceutical solvates and hydrates. Nevertheless, its content may be better suited for publication in journals focused on green chemistry and sustainability.

The authors have clearly discussed the limitations of the proposed method, including the need to prepare solvent mixtures under precise controlled conditions. Considering this, the following points should be addressed before resubmission of the manuscript:

- 1 – Please, check and correct the reaction scheme (there is an interrogation mark) presented in Figure 6 of page 14.
- 2 – It is important not only to investigate the formation of hydrates and solvates, but also to evaluate their stability under shelf-conditions and after incorporation into a formulation. Did the authors perform any physical stability study to the isolated hydrates and solvates?
- 3 – Considering the critical role of precise controlled conditions "...the experimental protocols need to be extremely carefully executed to avoid unexpected conversions" (page 17), did the authors perform the experiments in triplicate to ensure the accuracy and reproducibility of the results?
- 4 – The authors stated, based on direct experimental observation (page 17), that no amorphous forms were obtained during the experiments. Was this observation further confirmed using differential scanning calorimetry (DSC)? In the PXRD data presented in the Supplementary Information (Figures S3, S4, S6 and S8), there appears to be evidence of either an amorphous component ("halo" underneath the Bragg peaks) or potential formation of nanocrystals. To distinguish whether the observed PXRD "halo" is due to nanocrystals or amorphization, complementary DSC analysis is necessary to confirm the absence of a glass transition temperature.
- 5 - Considering that degradation can also occur during milling, did the authors check that no degradation occurred during ball milling?
- 6 – Please review the reference list and ensure consistency in the use of journal names. Use either the full journal name or its standard abbreviation.

Reviewer #3

(Remarks to the Author)

The paper *Shaken, not Stirred!* is interesting for the fact that it provides a quick method to determine solvate forming propensities of APIs in the presence of solvents. It is also shown that depending on the solvent concentration, different solvates can be produced. All results obtained by grinding have been verified by crystallisations in slurries confirming all outcomes and demonstrating that grinding or mechanochemistry is the more rapid method.

There is one important problem in the paper. Solvent activities have been calculated from concentrations, however, the assumptions for the conversions are incorrect. Solvent activities in the mixture are calculated using the assumption of a mixture of two pure solvents and activities are obtained from the NRTL model based among others on vaporisation data of the solvent mixtures. If the solvents would have been the only constituents of the system, these activities are correct, however, APIs are present within the system too. Most of the API is present in the solid state, which could be interpreted in terms of an activity of 1, but the fact that hydrates or solvates are forming is absolute proof that the approximations of mostly pure solvent (all eqs. in Figure 1) is not valid as interactions between solvent and API are undeniably present and thus the activity of the solvent is affected by the presence of the API. Solvents in solvates possess a well-defined activity, which is equal to the vapour pressure of the solvent above the solvate. The authors are very aware that impurities affect solvent activity as they mention the importance to avoid the presence of water in the solvents, however, they ignore the presence of the API "impurity". Activities change in the presence of the APIs, I suggest therefore to use solvent concentrations and not activities to report the results, concentrations are undeniable.

The reason that slurries and grinding give rise to comparable results with similar "activities" is because the conditions under which the solvates form are very similar. In both cases the system in equilibrium will contain saturated solutions of API; thus, solvents with the exact same activities (in grinding and slurry experiments) tending towards equilibrium. Except, these activities are definitely not the ones calculated by NRTL assuming pure solvents.

A related issue can be found in Figure 7. This figure appears to be or is even presented as a phase diagram, but it is not because one of the constituents, the API, is missing in the diagram. There should in fact be a fourth axis representing the API and one would expect the current drawing to represent an isopleth of the four-dimensional phase diagram. But even this is not the case, as in fact, the API concentration most likely changes with the saturation in the different solvent mixtures, so what we are looking at is most likely some curved surface in the quaternary system ($T = \sim 25^\circ\text{C}$). Therefore figure 7 is not a ternary solvent system, but a quaternary system in which the effect of the three solvents on the API under saturation (against hydrate/solvate/pure API..., surely all different API saturated concentrations) is demonstrated. It is far from a phase diagram even if this map may be of interest to crystal engineering. Thus this figure should be clearly described in what it represents and the term phase diagram should be avoided as this will cause confusion.

Some minor issues

>The objective of the figure just below the abstract (TOC figure?) is not clear and how is the repetition of the grinding jar with and without laboratory background relevant? This figure does not capture in its present state the core of the paper.

Page 9 top: refers to 4-hydroxybenzamide hemihydrate form I.

>chemical name is misspelled

Section 2.6

>The entire discussion remains rather vague. What is meant with $\alpha(\text{H}_2\text{O}/\text{DMF}) \approx 5$? is this the ratio between two activities? then write this explicitly with two alphas. Otherwise, what is it?

This discussion of should be clarified and indications as "very close" (4th line from the bottom of section 2.6) should be quantified. Activities run from 0 to 1, so differences of 0.1 are differences of 10% and are not necessarily 'close'. So be careful what is stated.

Figure caption 6, last line: Plots also show solid form transitions

>solid should be solid

Page 18, second paragraph, second line: it is very important to apply especial care in preparing

>especial - special

page 18, third paragraph, end: For scenario C, studying competing solvates work, we found that working with 5 μL per mg of API solids (rather than 2 μL per mg) helped reduce these uncertainties and led to observed activity ratios very close to those predicted from the critical activity values of the independent equilibria.

>This sentence is not clear, what is very close? and how exactly are conditions predicted from the critical activity values. Prediction can be understood on many different levels. This refers to section 2.6? Please be more clear and explicit.

Conclusions, first paragraph: By precisely controlling solvent activities, CSA-LAG enables the construction of complex phase diagrams across a range of solvation scenarios—including competitive solvation—with exceptional efficiency and accuracy.

>Conditions for the formation of solvates have been elucidated, but phase diagrams have not been produced or provided in this paper.

Conclusions, second paragraph: This approach identifies solvent activity thresholds that govern solvate formation, ...

>The paper identifies solvent concentration thresholds, the activities are not properly defined

>The caption of Figure F2 is not clear

Version 1:

Reviewer comments:

Reviewer #1

(Remarks to the Author)

The authors have satisfactorily addressed all the comments.

Reviewer #2

(Remarks to the Author)

The manuscript "Shaken, not stirred! Phase diagrams of Pharmaceutical Solvates", by Cruz-Cabeza et. al., presents a novel and sustainable mechanochemistry-based method for screening the formation of solvates and hydrates of pharmaceutical compounds, utilizing the solvent activity of solvent mixtures. The authors have thoroughly addressed all of my previous comments and have significantly improved the revised version of the manuscript. The current version is well-written, and the main message is clearly conveyed, making it accessible to a broad scientific audience.

Reviewer #3

(Remarks to the Author)

Unfortunately, the paper has not been sufficiently improved to justify publication.

The authors state that moving from concentrations to activities is one of the novelties of their work; however, so far the authors have not convincingly demonstrated that using estimated solvent activities instead of solvent ratios provides an added value and leads to new insights.

1. As presented in the paper, the estimated solvent activities are nothing more than a more complicated way of representing the solvent ratios used.

2. The authors state that they can neglect the effect of the API on the solvent and that this is common practice citing two papers from the literature.

- The paper by Firaha et al. in fact does take into account API activity as they base their analysis on the measured water vapour pressure of hydrates. This vapour pressure is therefore clearly dependent on the interaction between water and the API. It does not support the approach of the authors.

- The paper by Khankari et al. states that water activity can be altered by changing its concentration in mixed solvents. This neither supports the tenet of the authors to use activity based on the vapour pressures of individual solvents in solvent mixtures while neglecting the presence of API.

3. The authors use liquid assisted grinding, which implies high local pressures with a massive excess of solid API in the presence of some liquid. Locally this may easily lead to high supersaturation, even if the system may relax to equilibrium shortly after. Saturation implies that the activity of the crystallising substance (the API or its solvate) is 1 in solution and for supersaturation the activity will be higher. Neglecting the effect of the API under these circumstances is not convincing from a thermodynamic point of view. It is the crystallising entity and therefore its activity is of importance.

4. Repeating what I stated in the original review: if a solvate is formed, there is necessarily an effect of the API on the solvent, the actual solvate formation (= interaction) proves this. Dilute solution approximations (solvent activity = 1) breaks down in such a case, because a solvate is not dilute.

Version 2:

Reviewer comments:

Reviewer #2

(Remarks to the Author)

After carefully reviewing the manuscript "Shaken, not Stirred! Phase Diagrams of Pharmaceutical Solvates from Mechanochemistry" by Cruz-Cabeza et al., alongside the comments from reviewer 3 and respective authors' detailed response, I conclude that two complementary perspectives on the thermodynamics of CSA-LAG experiments are presented. Reviewer 3 correctly notes that local supersaturation and transient non-equilibrium states during LAG may influence the nucleation kinetics of the APIs. While this observation is valid at the microscopic or mechanistic level, the approach presented by the authors remains thermodynamically sound for mapping solvate and hydrate equilibria. In the experiments presented by the authors, the API is present in large excess and has a very low solubility, such that its activity in solution is effectively constant and negligible. Consequently, the equilibria are governed primarily by the chemical potential of the solvent, which is appropriately described using experimentally determined activities for real, non-ideal mixtures, where activity coefficients deviate from unity and therefore Raoult's law is no longer applied. Furthermore, the authors have provided empirical evidence confirming that the presence of dissolved API perturbs the

solvent activity by an amount smaller than the experimental uncertainty. Therefore, the use of solvent activities, rather than simple concentration ratios, is a valid and necessary thermodynamic approach for defining solvate and hydrate formation boundaries. In my opinion, the concerns raised by reviewer 3 pertain mainly to local kinetic effects, which do not undermine the equilibrium-based rationale of the methodology presented by the authors. Overall, the authors have addressed all comments comprehensively and have further improved the clarity of the manuscript.

Reviewer #3

(Remarks to the Author)

The article on Controlled solvent activity liquid assisted grinding clearly improved since the first version; however, the authors do not sufficiently take into account a possibly breakdown of one of their main assumptions: 'that solubility is low and that therefore the effect of the API on the activity of the solvent can be neglected'. They state that this is common practice, while citing numerous papers.

First of all, I agree with the authors that activity (or the chemical potential) is the quantity of importance that controls phase behavior in the system. Where my point of view deviates from that of the authors is to what extent the presence of API and the LAG process itself may change the activity of the solvent and that therefore the activity listed by the authors is not necessarily correct. In physical chemistry textbooks such as the one written by Peter Atkins, it is clearly stated that the presence of a solute will affect the chemical potential of the liquid and vice-versa (p 140 Eight edition of Physical Chemistry by Peter Atkins and Julio de Paula). Moreover, the activity coefficient of the solvent changes as a function of the concentration of the solute and vice versa (Margulles equations)(p162 Eight edition of Physical Chemistry by Peter Atkins and Julio de Paula)

In dilute solutions, the presence of a solute may be neglected and I am aware that this approximation is being used in the pharmaceutical literature for slurries for example. When those assumptions break down will depend on the system under study and should in fact be tested. In particular, if one intends to provide accurate data based on activities.

My problem is that in this paper, the authors study liquid assisted grinding, while using approximations that have been accepted for different conditions, but not for LAG, because the combination of LAG and activities is new as the authors state.

In the paper it is stated that the liquid-solid ratio is 2 parts of liquid for 1 part of solid in terms of weight. This is not a dilute system. It implies that most solvent will be in contact with a solid surface, even more so during the grinding process. There will be an extremely large interface, while the calculated activity coefficients of the solvents are based on bulk mixtures with relatively little surface or interface. Even if the assumption that the solution is dilute due to low API solubility would be true, this is not necessarily sufficient; the system in LAG consists mainly of interface. The assumption that the presence of the (solid) solute can be neglected should be validated, which is not done in this paper.

Moreover, the authors state themselves:

p21, l22

"We encountered this issue for the 4ohbzm system in CSA-LAG with ace:eg. The high solubility of both the anhydrous and acetone solvate in the liquid mixture meant that CSA-LAG could only be performed by lowering the liquid to solids ratio to 1 μ L/mg which in turn led to a window of activities for which mixtures of the anhydrous form and the acetone solvate were observed."

Here, the argument that the API exhibits low solubility is clearly not true.

The authors state that using activities and linking those to the observed phase behavior (solvate formation) is much better than only relying on concentrations. I agree with the authors, but I disagree with the fact that the authors control the activity during the grinding process.

What the authors have demonstrated in the paper is that their results coincide with those obtained by slurry under similar activities; however, how would the straight-line relationship look like in terms of liquid-liquid concentrations?

The LAG process is much faster than slurries, which is absolutely of interest, but they have not demonstrated that within the LAG process, the critical solvent activity is the same for a given solvate while varying the solvent mixtures and keeping the activity of the active solvent the same.

Some of the statements in the paper are too strong if based on the current results and should be written in a less definite form.

p3, l21 The novelty of CSA-LAG lies on the use of solvent mixtures of precise compositions which provides the ability to precisely tune solvent activities in the milling environment.

Due to the grinding process, the extremely large interface, and in certain cases high solubility, "precisely" is not accurate here. It should be removed.

p8, l14 This is a well-established simplification in pharmaceutical hydrate research for non-electrolyte systems, supported by decades of literature and experimental validation.

This may well be, but that doesn't mean that this simplification or assumption is valid for the current approach. It has not been convincingly established in this paper for this process.

p8, l19 Industrial crystallisation guidance explicitly demonstrates the use of NRTL binary solvent activity calculations for controlling water activity during hydrate crystallisations, again without including API effects in the activity calculations.

Indeed, however, the guidance does not include new approaches such as LAG. The working hypothesis should be tested.

p8, l23 While water activities can be measured directly (e.g., using a calibrated hygrometer), activities of other solvents cannot be experimentally determined.

This is not true and should be removed. Vapor pressures can be measured and they are a direct representation of the Gibbs energy, otherwise the NRTL calculations carried out by the authors would not be possible either.

p8, l24 As a result, the thermodynamic solvation behaviour of non-aqueous solvents has historically been overlooked, with solvates often discovered serendipitously rather than through systematic investigation.

As the previous statement is incorrect, it cannot be the reason for the serendipitous discovery of solvates. It is more likely that the interest is less, because solvates in APIs are something to avoid, not to aim for. Water is so relevant, because it is all present and acts often as a solvent for the API in the body. The sentence is misleading at the least and had better be removed.

p22, l1

For example, as shown in Table 2, the critical water activity for the NF-MH-I transition, measured using CSA-LAG in a h₂O:eg solvent mixture, is approximately 0.5—corresponding to a water mole fraction of 0.52 in that system. This activity threshold defines the thermodynamic limit for hydrate formation and remains consistent across solvent mixtures and experimental methods, thereby enabling robust prediction of behaviour. If nf is developed in its anhydrous form using a h₂O:ace mixture, maintaining water activity below 0.5 ensures the stability of the anhydrate; however, relying on mole fraction alone (e.g., 0.52) would be misleading, as this corresponds to a water activity of 0.76 in h₂O:ace, promoting hydrate formation.

I agree with the reasoning here, but it lacks proof. Why did the authors not show three different solvent mixtures in which the critical water activity is 'calculated' to be the same.

Right now, the paper demonstrates that LAG is very efficient in creating solvates, but not that the activity is controlled.

The authors use an approximation. It is their task to demonstrate that the approximation is valid within their system. It is not enough to state that it is common practice under other experimental settings. The authors should mention the shortcoming that the premise of controlled activities within LAG has not yet been fully demonstrated. If this is mentioned in the paper and the textual changes mentioned above have been implemented, this paper can be published.

This document provides a point by point response to reviewers with the authors' answers in blue. New text which has been added to the manuscript is provided in red and a marked revision of the main manuscript is provided.

=====

Reviewer #1 (Remarks to the Author):

This is a very interesting article that opens the door to numerous similar studies and, hopefully, encourages greater use of mechanochemistry as a screening tool, which is more sustainable than slurry methods. I greatly appreciate the article's ability to synthesize complex information—a task that is certainly not easy—both in textual form and through the use of equations and graphical representations.

Thank you for such positive feedback, we strongly believe in mechanochemistry as an excellent screening tool for solvates.

However, in my opinion, the manuscript would benefit from providing clearer guidance to the reader regarding the rationale for the choice of solvents and the four APIs, as well as for the selection of their solid forms.

We thank Reviewer 1 for this valuable suggestion. We have elaborated on this and the choice of APIs has been clarified in pages 9-10 as part of the Systems section. We state:

“Since we wanted to show the generality and universality of CSA-LAG, four APIs of various molecular complexities were selected for our experimental studies (Figure 3a), namely theophylline (theo), 4-hydroxybenzamide (4ohbzm), carbamazepine (cbz) and nitrofurantoin (nf). These four systems cover from small rigid (theo) or medium partially rigid compounds (cbz) to small lightly flexible (4ohbzm) and larger moderately flexible compounds (nf). The systems were also chosen because they are all known to form hydrates of diverse stoichiometries (which we wanted to explore) as well as a variety of other solvates of various nature (Table 1). The diverse nature and solid form richness of these APIs, therefore, make them ideal to explore CSA-LAG across a diverse range of molecular and solvate systems.”

The choice of solvates for the mixture has also been further explained in the same section. We have added the following text to fully clarify the solvent choices:

“The choice of solvents for the generation of the mixtures was made based on two distinct conditions: a) one solvent requires to be active (e.g. leading to hydration/solvation) and b) a second solvent is required to modulate the activity of the first through compositional variations (hence full miscibility with the active solvent is required) and, for simplicity, this second solvent is typically chosen to be inactive. The choice of the active solvent is dictated by the hydrate or solvate of interest. We studied four diverse active solvents, namely water, dmsol, dmf and acetone (Table 1). The choice of inactive solvent was informed by the literature: it is simply a solvent leading to crystals of the unsolvated API under study. Common passive solvents used in crystallisation are ethanol and acetone. Alternatively, passive solvents can be explored by LAG experiments. Ethylene glycol was also selected since it is usually passive and has a high boiling point – thus solvent evaporation is minimised. All binary mixtures used for each system are given in Table 1, details of their generation in the ESI (sections

1.6 and 1.7) and activity against composition behaviour shown for two of the mixtures in Figure 3b (**h₂o:etoh** and the **dmf:etoh**). The dependence of the activity on molar composition in the mixtures is highly system dependent and can significantly deviate from linearity and Raoult's law.”

It also lacks an early explanation of the role of the solvating solvent and the inactive carrier. The role of the solvent, including as a solvating solvent in LAG, remains insufficiently understood in the scientific literature, which makes it even more important to explain it clearly here—particularly if the authors are confident in their interpretation.

In summary, it would be valuable to include a concise introductory section that sets the stage before presenting the core of the work. This would help guide the reader more smoothly through the subsequent sections.

Thank you for this. We have added an explanation on the role of solvent in LAG in the introduction given our knowledge and some recent works. The addition to this in the introduction together with the explanations in section 2.2, should hopefully clarify the choices of solvent, and allow the extrapolation to extended solvate systems and solvents. The paragraph in the introduction reads:

“In 2002, Shan et al. first reported that the formation of cocrystals via ball-mill grinding could be significantly accelerated by adding “minor amounts of an appropriate solvent” to the mill, introducing LAG³⁷. Early works with LAG were referred to as “solvent-drop grinding” given that, experimentally, only a few drops of solvent were added to the milling experiments³⁸. In 2004 it was shown that LAG with different solvents can lead to different polymorphs³⁹, and in 2016 that those outcomes are a consequence of crystal size effects³². **The fundamental role of the solvent in the LAG interconversion reactions is still not fully understood. Recent population balance modelling work has shown that crystal breakage, crystal growth and Oswald ripening processes all occur in the mill and that the solvent presence can impact this delicate balance and hence the resulting outcomes²⁶. Crucially, the solvent nature and amount (relative to solids) impact the rate of crystal growth of the new phases in the mill, with larger amounts of solvent and solvents with higher solubility leading to larger crystals^{26,40}.”**

The discussion does not adequately clarify the theoretical basis for the rationalization of the three scenarios in the equations. As a result, the text delves straight into the subject matter without providing an adequate introductory section to guide the reader. As it stands, the article remains rather dense and could be challenging to follow without this context.

We thank Reviewer 1 for this valuable feedback. We agree that the previous version of the manuscript potentially moved too quickly into the equations without sufficient introductory framing, and rationalisation behind the selected solvation scenaria. To address this, we have now added an introductory paragraph at the beginning of Section 2.1. This paragraph reads:

“Solvate formation can be interpreted within the broader framework of phase equilibria, wherein the relative stability of solid forms is governed by the chemical potential⁴⁸. At thermodynamic equilibrium, the chemical potential of each component remains uniform across all coexisting solid and liquid phases. Deviations from equilibrium introduce a driving force for solvation or desolvation, determined by the solvent activity in the liquid phase⁴⁹. Within the context of CSA-LAG, these thermodynamic principles can be adapted to describe relevant mechanochemical transformations, enabling the expression of solvate equilibria in terms of equilibrium constants, critical activities, and changes in free energy. In our framework nomenclature we use A for the API, S and T for different solvents, m , n and w for different solvent to API stoichiometries in the solid-state, (s) and (l) to indicate the solid and liquid states, K for the equilibrium constant, Q for the reaction quotient and α for the activities. For simplicity, the stoichiometry of the API in the solid-state is considered as 1 relative to m , n and w ; inclusion of a different stoichiometry for the API in the formulations is straightforward. From this foundation, three generalised scenarios emerge that capture most common solid-state solvation phenomena: (a) single solvation, (b) stepwise solvation, and (c) competing solvation. These scenarios are illustrated in Figure 1 and serve as the theoretical basis for the subsequent analysis. Scenario (a), the most prevalent, involves the transformation of a neat API into a single solvate of defined stoichiometry. Scenario (b), while less common, is frequently encountered in hydrate systems, where the API may exist in neat form and as different solvate structures with the same solvent but different stoichiometries. Scenario (c), which is comparatively rare, addresses the thermodynamic competition between two different types of solvates of the same API. Although additional scenarios can be envisaged—such as competition among three solvents for solid state solvation, multi-step solvation involving three or more stoichiometries, or the formation of heterosolvates—, these are significantly less common so they will not be presented here and can be derived with analogous thermodynamic formulations as those established here.”

Finally, while the work is interesting, the overall approach is not entirely novel, which further highlights the importance of emphasizing what is truly new in this study.

We thank Reviewer 1 for this observation and to some extent agree that while both slurries and mechanochemical experiments have long been used to investigate hydrate and solvate formation, the novelty in our work lies in the development and demonstration of CSA-LAG as a systematic, transferable and quantitative framework for controlling and probing solvent activities during mechanochemical screening.

The following is novel:

- **Precise control of solvent activities** via the use of solvent mixtures.
- **Downscaled, rapid equilibration** enabled by mechanochemistry.

- **Coupling** of solution thermodynamics, crystal forms stabilities and mechanochemistry
- **Proof of CSA-LAG generality across multiple APIs** and solvents.

We clarify the novelty at the end in the introduction, which now reads:

“Here we present the framework of thermodynamic equations that rationalise solvate formation in the mill and develop a robust experimental method for LAG under controlled-solvent activity conditions (CSA-LAG). **The novelty of CSA-LAG lies on the use of solvent mixtures of precise compositions which provides the ability to precisely tune solvent activities in the milling environment. The solvent mixtures contain of at least one active solvent (able to solvate the system in the solid-state) and a passive solvent also referred to as the carrier solvent (used to tune the composition of the mixture and the activity of the active solvent).** CSA-LAG allows for the accurate and rapid exploration of phase diagrams of solvates and hydrates, including systems with multiple stoichiometries and competing solvates. We show the robustness, accuracy **and generality** of CSA-LAG by exploring multiple solvate maps for four compounds of pharmaceutical interest **at room temperature.**”

Suggestions for Improvement

- **Clarify Experimental Rationale:** The article would benefit from more guidance regarding the selection of solvents, APIs, and their solid forms. An early explanation of the role of the solvating solvent and the inactive carrier is missing, and its absence makes the manuscript harder to follow. A brief introductory section outlining the rationale behind these choices would greatly improve readability and accessibility.

The choice of APIs and their solid forms has been clarified further in the Systems section. As explained above.

- **Process Conditions and Temperature Control:** The process conditions are not clearly specified, and the claim that the temperature remains at 25 °C during milling (30 Hz for 30–45 minutes) seems implausible. Mechanochemical reactions are highly sensitive to temperature and other process variables. More precise measurement or control of these parameters is essential. The authors should provide clarification and, ideally, include temperature monitoring data.

Thanks for this. We discussed this in the “Discussion Section” but this is clearly too late. We precool jars to 20 Celsius since we know (through temperature monitoring) that this gives jar temperatures of 24-26 °C after milling for 30 mins in our Retsch MM400 mills. So we have now included this in Figure 2 to make sure

that is clearly shown in the methodology and explained the rationale in the same section. We have added the paragraph below to section 2 and stated temperature along with all phase diagrams presented. We further provide temperature monitoring and evidence for the effectiveness of the precooling methodology on achieving final temperatures within our effective working temperature window, as seen in tables S13 and S14 of Section 2.4 of the ESI.

“Temperature is another critical factor in these equilibria. Our current milling setup does not allow for active temperature control. To address this, jars and milling balls were pre-cooled to 20 °C prior to milling. We observed that milling raises the jar temperature by approximately 5 °C (measured ex situ using an infrared thermometer), hence we achieve an effective milling temperature of ~25 °C after precooling to 20 °C (ESI, section 2.4) — for consistency with the temperature of the VLE data used for activity calculations. Future iterations of the setup will incorporate more precise temperature control. In the absence of such control, we recommend monitoring temperature together with appropriate precooling.”

- Consistency Between Main Text and Supplementary Information: The Supplementary Information (particularly section 1.3) contains discrepancies in milling time, powder quantity, and liquid volume compared to the main text. These differences should be addressed, as they compromise reproducibility. Moreover, the gravimetric solubility measurements lack key details—most notably, the temperature at which they were conducted. This omission is critical, given the temperature dependence of solubility.

We thank Reviewer 1 for their comment. There were many inconsistencies in the ESI so we really appreciate this being raised. We have significantly improved the presentation of the ESI and made sure everything was clear and consistent with the presentation of data in the main manuscript. This has now improved enormously. Thank you.

- Graphical Abstract and Data Presentation: The graphical abstract does not effectively convey the study’s focus. I recommend revising it to better reflect the core findings, maybe starting from the cscheme in Figure 2.

Thank you. We have significantly improved the graphical abstract, we hope it conveys the message better.

Additionally, a summary table listing the APIs, solvents, and solvent mixtures used—including composition ranges for binary and ternary systems—would be extremely helpful for readers navigating the experimental design.

Table 1 now contains all this information. Thank you.

- Structural Clarity: The Supplementary Information includes a well-organized index that could be partially integrated into the main text to improve the logical flow. This

would help guide readers through the dense content more effectively.

Thank you for this. We are not 100% sure as to what the reviewer is requesting here. Apologies. We think the reviewer would like us to cite the different ESI sections within the main manuscript. We have done that now extensively.

OTHER SUGGESTIONS:

- A more comprehensive discussion of the relevant literature would have been also appreciated, rather than treating it only briefly. For example: Cruz-Cabeza, A. J.; Wright, S. E.; Bacchi, A. On the Entropy Cost of Making Solvates. *Chem. Commun.* 2020, 56 (38), 5127– 5130, DOI: 10.1039/D0CC01050B; Conformational Change in Molecular Crystals: Impact of Solvate Formation and Importance of Conformational Free Energies DOI: 10.1021/acs.cgd.1c00833; Lombard, J.; Laker, H.; Prins, F.; Wahl, H.; le Roex, T.; Haynes, D. A. Selectivity of Hosts for Guests by Liquid-Assisted Grinding: Differences between Solution and Mechanochemistry. *CrystEngComm* 2021, 23 (42), 7380– 7384, DOI: 10.1039/D1CE01286J

Thank you. The first two references have been added to the introduction and the last one to the discussions in addressing the point regarding the potential for different outcomes between LAG and solution experiments (see below). Regarding the first two references in the introduction, this has been added:

“In the early 2000s, the Jones group conducted several studies demonstrating that pharmaceutical hydrates³⁹ and solvates⁴⁰ could be generated using simple LAG. The resulting LAG phases were predicted computationally through lattice energy calculations³², with entropic⁴¹ and conformational effects being incorporated into such models more recently⁴².”

- A table summarizing all the boundaries in both mechanochemical LAG and slurry experiments is strongly recommended.

Thank you for this. The boundaries for scenario (a) are shown summarised in table 2. Scenarios (b) and (c) each study a case at a time and are all summarised in figures 5 and 6. We feel this presentation of data is effective. Tables for all data are also in the ESI and referred to in the main text. It is difficult to summarise all data in a single table when you start mixing scenarios and the data of the different scenarios is already summarised in Table2.

- The observation of results identical to those obtained by slurry is interesting; however, based on my experience, this is not very common. In fact, certain solvates formed mechanochemically do not typically form via slurry methods. I would encourage the authors to emphasize this distinction and discuss the implications of this finding in greater detail.

Discrepancies of this kind are usually observed if crystallisation (rather than slurries) is used or the slurries are not allowed sufficient time to reach equilibrium. We have added the following to the beginning of the discussion and also included the reference by Haynes et al. (*CrystEngComm*) and a reference to Bucar et al. The paragraph now reads:

“We have shown that CSA-LAG is a fast, reliable and robust method to determine boundaries of solvate formation in single, sequential and competing solvation reactions. The agreement with the boundaries found with more “traditional” slurry experiments for all systems studied here is excellent as shown from the linear correlation in Figure 8 with R^2 factor of 0.975. The excellent agreement is unsurprising since in both sets of experimentations we have driven the systems to thermodynamic equilibrium. We note, however, that some literature exists reporting disagreements of outcomes between solution and solid-state methods^{46,52} for cocrystals as well as host-guest systems and solvates. Our interpretation of these disagreements is that kinetic effects are likely at play more significantly in synthesis methods from solution where equilibration is not achieved (simple crystallisation rather than slurries). In our own experience working with solvates, we have sometimes obtained metastable solvates from solution crystallisation which cannot be obtained by CSA-LAG and we have also produced stable solvates from CSA-LAG that cannot be produced from solution (even slurries) due to difficulties in nucleation. The fact that thermodynamic equilibrium is rapidly and consistently achieved via LAG, naturally makes CSA-LAG a more robust methodology for phase diagram generation than solution methods.”

- This table is not clear: Table S7 — Ternary molar composition of DMF, water, and ethanol used to prepare mixtures of known initial solvent activity. Please clarify the meaning of all acronyms used, either directly in the table or in a footnote, to ensure that the information is readily understandable to the reader.

Thank you for this. This table S7 has been removed. It is not useful and nor helpful. The models used with ASPEN are already described in the methods and the sub details of these models are accessible through the software.

- Rietveld refinement is not mentioned in the Materials and Methods section, but only in the Supplementary Information and in the Results. It should be described in sufficient detail in the Materials and Methods in the main paper.

This has been added to the main article too. Thank you.

- Is it correct to understand that no mixtures of solid forms are obtained at any given solvent activity? From Tables S13...it seems that only single solid forms are reported. Clarification on whether solid form mixtures occur—or why they do not—would be valuable for interpreting the results accurately. Although this is reported for scenario C, it is quite remarkable that in the other cases no sample appears to deviate from ideality. Given that there is often a discrepancy between theory and practice, this point would deserve further discussion.

Thank you. The observation of mixtures is now also discussed in the discussion and should be clear. It now reads:

“Over the course of the CSA-LAG experiments carried out in our study, mixtures of phases were only observed under a small number of milling conditions (ESI, section 2). There are three important considerations that account for these mixtures. First, if one of the liquid-mixtures used for the LAG has a solvent activity very close to the critical value, it is possible that the solids at equilibrium contain both solvated and unsolvated phases, since both are equally stable at the critical activity. Second, we have used a liquid to solids ratio of 2 $\mu\text{L}/\text{mg}$ which means that the liquid is in overall molar excess relative to the solids by about a factor of 16 to 30 (depending on the system). However, as solvent mixtures are used and composition of the active solvent and its activity is progressively increased, there exists a small window of activities where the number of mols of active solvent fall slightly below the stoichiometric requirement for solvation, which can lead to mixed phases. This is rare since the windows of activities where this may occur more commonly will lie between 0-0.2 in most cases. In the case of competing solvation (scenario C) this was overcome by using a very high liquid to solids ratio of 5 $\mu\text{L}/\text{mg}$ bypassing stoichiometry limitations. Third, for instances where the API is very soluble in both the active and the carrier solvent, the liquid to solid ratio needs to be reduced in order to avoid full dissolution. These conditions will lead to wider windows where the active solvent to API stoichiometries will lie below the stoichiometric reaction needs. We encountered this issue for the 4ohbzm system in CSA-LAG with ace:eg. The high solubility of both the anhydrous and acetone solvate in the liquid mixture meant that CSA-LAG could only be performed by lowering the solids to liquid ratio to 1 $\mu\text{L}/\text{mg}$ which in turn led to a window of activities for which mixtures of the anhydrous form and the acetone solvate were observed. The critical activity always lies between the last activity that gives the pure anhydrous form and the first activity that leads to solvation (independent on whether the outcome is the pure solvate or a mixture).”

- The table listing Crystal Form 4OHBZM-ACET-I and NF-DMF-II, along with their crystallographic data and CCDC deposition numbers (2443101 and 2443102), appears to report structural details of newly resolved crystals rather than an overview of thermodynamic boundaries as indicated by the title of Table S28 Table S 28: Overview of the thermodynamic boundaries of the hydrates and solvates studies with CSA-LAG and slurring. Please clarify the intended content and labeling of this table to avoid confusion.

Thank you for spotting this mistake. This table’s heading is now updated, along with a more concise summary of the crystallographic information. We have significantly improved the ESI.

- ABSTRACT: Scientific acronyms are quite fashionable these days — it seems every researcher needs to be remembered by one: “I was the one who introduced this technique” CSA-LAG... well, I’m not entirely convinced. Perhaps an acronym with a smoother flow could be found. Of course, this is a matter of personal preference.

AI suggested to me:

GRASP – Grinding under Regulated Activity of Solvent Phase

SAM – Solvent Activity Milling

SAGE – Solvent Activity Guided Experiment

SAGE-M – Solvent Activity Guided Experiment – Milling

SARM – Solvent Activity Regulated Milling

Thank you for the suggestions. We use an acronym for the technique for simplicity and we needed one in order to make the text flow. We would like to keep CSA-LAG. We spent a significant amount of time trying to agree on this, and debated it with the entire team. “Controlled Solvent Activity LAG” describes best what we have done hence CSA-LAG is our preferred acronym and we would like to keep it. LAG is well established, so it is a variant of very well established and widely accepted LAG and we think LAG shall be in the name.

- LINE 32 It is not clear to me — are there two graphical abstracts? In my opinion, the concept of CSA is not clearly highlighted, nor is the motion of the jars effectively represented. A modification of Figure 1 that continues the storyline beyond the PXRD analysis — showing the construction of the phase diagram and the application of the three-scenario equations — would be an improvement.

Thank you. The two figures are a graphical abstract and a cover art. The graphical abstract has now been improved to better tell the story in a minimal figure. We think it is an improvement. The cover art is just a beautiful art creation that does not necessarily tell the details and we wish the journal to consider for cover (it may not even be published, it is just a suggestion for cover).

- LINE 79 I would suggest including this article as well: <https://dx.doi.org/10.3390/cryst14040374>. Although it was published in a journal that is not particularly prestigious, given the limited literature available on this topic, I believe its inclusion is appropriate.

We thank Reviewer 1 for bringing this article to our attention, we see the relevance in the work and have now added this to the introduction.

- LINE 94: The meaning of m , n , ... w ... in relation to the stoichiometry is not entirely clear. Given that solvent mixtures are being added, it is unclear how these variables are defined and what their specific stoichiometric significance is in this context.

Those stoichiometries are in the solid-state. We have now clarified this further by adding: “In our nomenclature we use A for the API, S and T for two different solvents, m , n and w for different stoichiometries in the solid-state.”

Some information can be taken from the ESI and included in the main.

We think the paper is already quite extensive, so would refrain from moving more data from the ESI into the main, unless there is something specific that needs further attention.

In my opinion, what remains particularly unclear is the stoichiometric ratio between the solid and the solvent, whether the solvent is used alone or in mixtures.

This lack of clarity is further compounded by the inconsistent reporting of the solid-to-liquid ratio throughout the manuscript. The experimental section in the main text (line 449 and following), the ESI (1.2. milling pag. 3), and the results discussion each refer to different ratios for the LAG conditions, making it difficult to assess the reproducibility and interpret the mechanistic implications of the solvate formation. A clear and consistent definition of the stoichiometry used is essential for evaluating the validity of the conclusions.

Thank you. We had made inconsistent reports across the ESI and the manuscript. We have now revised this to make sure it has been appropriately reported. We used 100 mg of solids to 200 microliters of liquid – except for the competing solvates caused because of the rationale behind it. We had a parallel set of work that used slightly different amounts, hence the confusion, but now we have revised our report and records to make sure we reported the correct values consistently. The liquid could be a pure solvent or a mixture of 2 or even 3 solvents. Always given at 100 mg of solids to 200 microliters. We have reiterated this in section 2.2 and added a comment on the implications of this to the observation of mixture of phases (see reply above). This shall all be consistent and clear.

- LINE 98 Does the equation really work for all solvates? Wouldn't it be more accurate to state that it "is expected" to work for all types of stoichiometric solvates? Does it really work also for mixed (hetero) stoichiometric solvates containing two solvents? If not, which equation would apply in that case?

Indeed, we have identified as well that these equations form the sole basis for the study of the solvation thermodynamics and in particular the formation of stoichiometric solvates as we have mentioned too in line 98.

Whilst heterosolvates do exist, their lack of applications, rarity and general avoidance lead us to not extend our work as far in this manuscript and potentially leave the exploration of heterosolvation thermodynamics in a follow-up paper. We noted this in section 2.1 clearly stating the exclusion of heterosolvates. The following sentence explains:

"Although additional scenarios can be envisaged —such as competition among three solvents for solid state solvation, multi-step solvation involving three or more stoichiometries, or the formation of heterosolvates—, these are significantly less

common so they will not be presented here and can be derived with analogous thermodynamic formulations as those established here.”

- LINE 175: this sentence must be written near Table 1. For a reader with expertise in pharmaceutical solid forms, the rationale for selecting these particular solid forms is not entirely clear, especially considering that multiple forms are known for each of these APIs

This has been further explained in Section 3 which provides a full rationale for this.

LINE 181: please note that this statement appears to be inconsistent with the choice of acetone

We thank Reviewer 1 for noticing such a detail. Our statement reflects the general principle that low volatility solvents reduce the compositional drift. We nevertheless chose to include a single example intentionally because i) acetone is a widely used polar aprotic solvent in pharmaceutical screening and crystallisation, ii) it expands the polarity space of our dataset and iii) several target systems are known to strongly respond to acetone water mixtures, also in several papers in literature we have spotted the use of acetone and methanol water mixtures.

Whilst we show that CSA-LAG can still handle case of volatile solvents, we strongly encourage readers to try to avoid such solvents, regardless of the screening method. This is the reason as to which we introduce Ethylene Glycol water mixtures, which are the excellent alternative to ethanol/acetone cases (assuming that ethylene glycol does not form a solvate with the API). Other benefits apart from low volatilities include a linear following of Raoult's law, making mixture preparation easier and less compositionally sensitive.

LINE 224 In my opinion, this sentence would be more appropriate in the Introduction section

Thank you. Indeed, this is now in the Systems section, where the choice of solvents is introduced.

LINE 256 In my opinion, this sentence would be more appropriate in the Introduction section.

We thank Reviewer 1 for this suggestion. In this case, we feel it helps the presentation of results so we would like to keep it there.

LINE 308 in water formula the subscript 2 is not visible

This is an issue of the conversion to the pdf. This will be resolved with the editorial office. Thank you.

LINE 334 Before linear regression, a table summarizing all the boundaries in both mechanochemical LAG and slurry experiments is strongly recommended. Please note that the ESI appears to contain an incorrect boundaries table, as it does not include the boundaries but the crystal parameters of the new solvates. So, in fact, the boundaries are not reported anywhere. It is essential that the correct data be provided to ensure transparency and allow for meaningful interpretation and replication of the results.

We have and do actually summarise in Table 2 all the studied equilibrium reactions studies both with CSA-LAG and slurries. This includes all the hydrate cases apart from 4OHBZM which is reserved for its standalone case in scenario B and all solvates. We are supportive of transparency and thus have included all the information on both active/passive solvents chosen for each case study and include a detailed description of those results in the ESI too, please refer to Section 2.3 for the data on the solvate boundaries along with their summary in Table 2 in the main manuscript.

FIGURE 8 : "solvant" instead of "solvent"

We thank Reviewer 1 for pointing out this typo, it has been amended in the manuscript.

LINE 357: Slurry are suspensions not solution.

Solution has been changed to slurry.

LINE 365: It is widely recognized that liquid-assisted grinding (LAG) typically does not yield amorphous forms. Therefore, this aspect does not seem surprising. However, it is somewhat surprising that no mixed solvates have been observed in this study. Clarification or discussion on this point would enhance the manuscript.

In our study we indeed did not detect the formation of mixed solvates or heterosolvates, under any solvent activity condition tested. The lack of the appearance of mixed solvates or heterosolvates is intentional and we strongly believe lies in the targeted selection of the active and passive solvents of the binary solvent mixture. As we have described in the manuscript the sole role of the active solvent is the solvation of the compound of interest, whilst

crystallisation and Liquid Assisted Grinding using the passive solvent should always lead to the desired unsolvated form.

These outcomes are consistent with our understanding that the systems examined preferentially and thermodynamically driven form discrete stoichiometric solvates other than hetero or mixed solvates.

We had considered the additions of a case of heterosolvation thermodynamics, although their rarity and general avoidance due to practical reason led us to not explore heterosolvation in this case.

LINE 442: please explain the choice of the 6 solvents

The selection of the six solvents was on the basis of commonly used solvents in crystallisation studies and screening. We have provided further insight on solvent selection in the Methods Section under Preparation of solvent mixtures. Throughout the manuscript we try to emphasize the generality of CSA-LAG to be used with any combination of miscible solvents provided thermodynamic modelling for those being available.

LINE 447 Please specify the container in which prepared solvent mixtures were stored and the type of lid used.

The solutions were prepared in screw thread clear borosilicate glass vials with a reduced neck, sealed vial phenolic caps and polyvinyl faced internal liners. Right after closing, the seal between the glass and the lid was wrapped with approximately 15cm of unstretched parafilm. This has been added to the methods section as requested.

LINE 450 Please specify the number of balls used and the type of material. From the graphical abstract, it appears that a single ball was used and that it is made of a different material than the jar.

A single ball was used per milling jar, we have now added this to the requested method section.

LINE 451: in the results section you stated 2 microliter for each microgram. it is contradictory with this sentence.

Fixed, this is 100 mg. Thanks.

LINE 453: here you mentioned 2-30 min milling time; in the Results section only 30 min is mentioned. Then in ESI 30-60 min. It is unfortunate to find such divergent experimental conditions reported for mechanochemical reactions, which are known to be highly sensitive to process variables. This is particularly concerning in a study that claims the temperature remains constant at 25 °C throughout. Without consistent and well-defined parameters, the reliability of the mechanistic conclusions is compromised.

We have clarified all parameters and the temperature too. This should be all consistent and clear through the manuscript. We apologise for the inconsistencies, which are now resolved. On the development of the manuscript, various ratios of solvents and times were used until we converged to those applied consistently throughout the work. We have rechecked everything and the values in the manuscript are now consistent.

LINE 457 Please specify how the temperature was measured — during the milling process or after opening the jar at the end. While it is true that short milling times, even at 30 Hz, may result in temperatures around 25 °C, in my personal experience and from literature data, milling for 45 minutes at 30 Hz significantly exceeds 25 °C and can reach approximately 35-40 °C.

We only measured the temperature on the jar. And we precooled the jar, as explained. In our case, the temperature after milling was between 24 and 26 Celsius. We have also addressed evidence of the precooling effect on the final temperatures in Tables S13-14 of the ESI Section 2.4. This now reads:

“Temperature is another critical factor in these equilibria. Our current milling setup does not allow for active temperature control. To address this, jars and milling balls were precooled to 20 °C prior to milling. We observed that milling raises the jar temperature by approximately 5 °C (measured ex situ using an infrared thermometer), hence we achieve an effective milling temperature of ~25 °C after precooling to 20 °C (ESI, section 2.4) — for consistency with the temperature of the VLE data used for activity calculations. Future iterations of the setup will incorporate more precise temperature control. In the absence of such control, we recommend monitoring temperature together with appropriate precooling.”

Line 457 Please specify the temperature. A similar lack of attention to experimental detail is evident in the gravimetric solubility measurements, where crucial parameters such as temperature are not even mentioned (see ESI section 1.3). Given the temperature dependence of solubility, this omission raises concerns about the reproducibility and accuracy of the reported values.

These details are now consistently added throughout. Thank you.

=====

Reviewer #2 (Remarks to the Author):

The manuscript “Shaken, not stirred! Phase diagrams of Pharmaceutical Solvates”, by Cruz-Cabeza et. al., describes a new method based on mechanochemistry for screening the formation of solvates and hydrates of pharmaceutical compounds using the solvent activity of solvent mixtures. Although the proposed method demonstrates advantages over slurry methods for screening new solvates and hydrates, mechanochemical methods, particularly ball mill grinding, are already well-established. These methods have been employed to screen the formation of both crystalline (e.g., polymorphs, co-crystals/salts) and amorphous forms of pharmaceutical compounds by varying conditions such as solvent type, ball mill type, milling time, frequency, and the number and size of milling balls.

The novelty of the present work lies on using the solvent activity of solvent mixtures to define the experimental conditions under which a compound will form a hydrate or solvate. While the method can be extended to other pharmaceutical compounds, a screening process remains necessary depending on the specific system studied.

Thank you for the positive feedback. The use of solvent activities is really novel and provide a robust framework able to predict and anticipate solvation in mechanochemical reactions.

Moreover, when scaling up the process, particularly for industrial applications using e.g., industrial-scale mills, the experimental conditions optimized for the studied model systems (theophylline, 4-hydroxybenzamide, carbamazepine, and nitrofurantoin) are likely to change. Even if a hydrate form is identified under specific laboratory-scale conditions, it may not be reproducible upon scale-up. Additionally, any adjustments to the scale-up process may lead to the formation of other solid-state forms. Did the authors consider varying conditions such as the amount of solvent (fixed here at 200 μL), milling time, and type of ball mill (e.g., using a planetary mill) to investigate how these factors influence the obtained results?

We thank Reviewer 2 for this insightful comment. We agree that scale-up of mechanochemistry to industrial or continuous mills is complex, with a lot of ongoing research investigating this and beyond the scope of our current study.

The focus of our work is to downscale and accelerate solvate screenings, and hence we have developed CSA-LAG. Since our methods and solvate boundaries are driven by thermodynamics and critical solvent activities are thermodynamic properties and hence intrinsic, and the scale of the milling experiment should not impact this. We added a paragraph on scalability to Section 2.2 which reads:

“In CSA-LAG, solids loading is a tuneable parameter that can be adjusted alongside liquid volume, milling times, and the number of milling balls. Provided the reactions reach equilibrium, the outcomes are governed by thermodynamics and remain independent of the absolute scale. To accommodate different experimental needs, increasing the solid quantity requires a proportional increase in liquid (we use an overall 0.5 mg/ μL ratio), and may also necessitate longer milling times or additional balls to

ensure equilibrium is achieved. An initial exploration of the variables is recommended if the loadings are to be altered prior to map generation.”

The work demonstrates potential for a more sustainable approach to screening pharmaceutical solvates and hydrates. Nevertheless, its content may be better suited for publication in journals focused on green chemistry and sustainability.

We strongly believe our paper fits very well in Nature Communications because of the novelty, generality and the wide applicability of our method. There is an increasing interest in addressing hydration and challenges it brings to pharmaceutical materials, as seen by the recently increasing Nature publications. Further to this, the topic is very much aligned with a recent papers published in nature:

Eaby, A.C., Myburgh, D.C., Kosimov, A. *et al.* Dehydration of a crystal hydrate at subglacial temperatures. *Nature* **616**, 288–292 (2023). <https://doi.org/10.1038/s41586-023-05749-7>

Firaha, D., Liu, Y.M., van de Streek, J. *et al.* Predicting crystal form stability under real-world conditions. *Nature* **623**, 324–328 (2023). <https://doi.org/10.1038/s41586-023-06587-3>

The authors have clearly discussed the limitations of the proposed method, including the need to prepare solvent mixtures under precise controlled conditions. Considering this, the following points should be addressed before resubmission of the manuscript:

1 – Please, check and correct the reaction scheme (there is an interrogation mark) presented in Figure 6 of page 14.

We apologise for this. It seems this was an issue related to the word to pdf conversion. This will be resolved with the editorial office.

2 – It is important not only to investigate the formation of hydrates and solvates, but also to evaluate their stability under shelf-conditions and after incorporation into a formulation. Did the authors perform any physical stability study to the isolated hydrates and solvates?

The physical stability of a form under shelf-conditions will be dictated by the environmental conditions and the phase maps, which we determine with CSA-LAG. Hence a solvate stored under conditions that favour the solvate will never change. If stored under conditions that lead to desolvation, it will change and the kinetics of this process can be very complex. The issue of studying such kinetics is beyond the scope of our current paper (focused on thermodynamics). But this is indeed something we will investigate in the near future.

3 – Considering the critical role of precise controlled conditions “...the experimental

protocols need to be extremely carefully executed to avoid unexpected conversions” (page 17), did the authors perform the experiments in triplicate to ensure the accuracy and reproducibility of the results?

The same experiment under the same conditions will always lead to the same outcome, since this outcome is driven by thermodynamics. In the above statement all we wanted to convey was that the protocol was carefully planned and executed to make sure that the conditions of the experiment are accurate. We are confident with the outcomes and as long as the protocol is executed carefully repetitions are not required. This is further supported by the excellent correlation we obtain between CSA-LAG and slurring derived values in Figure 8 (R^2 of 0.975). To reiterate this we have rephrased the sentence to:

“Since the stability of solvates is intricately dependent on environmental conditions, the experimental protocols need to be carefully executed to avoid **unexpected changes in the liquid compositions leading to changes in activities and unexplained conversions.**”

4 – The authors stated, based on direct experimental observation (page 17), that no amorphous forms were obtained during the experiments. Was this observation further confirmed using differential scanning calorimetry (DSC)? In the PXRD data presented in the Supplementary Information (Figures S3, S4, S6 and S8), there appears to be evidence of either an amorphous component ("halo" underneath the Bragg peaks) or potential formation of nanocrystals. To distinguish whether the observed PXRD "halo" is due to nanocrystals or amorphization, complementary DSC analysis is necessary to confirm the absence of a glass transition temperature.

We thank Reviewer 2 for pointing this and the opportunity to clarify and improve the manuscript. We would like to clarify that the previous Figures S2, S3, S6 and S8 depict the resulting powder diffraction patterns of background un-subtracted (Kapton plus Si wafer) scans. The artificial "halo" which is responsible for the increased background is now clearly addressed in the ESI Section 1.8, where we illustrate that the background is due to the Kapton tape not to amorphisation. We now also illustrate all of our PXRD overlays with the Kapton contributions now subtracted, as seen in the ESI Figures S6 and S7. DSC is of difficult use in our case, because the samples are wet and the interpretation would be very difficult. However, we observe very nice diffraction peaks consistent with samples of good crystallinity as well as we know from prior work that amorphization with LAG is extremely uniquely. Further to that, our ability to perform quantitative phase analysis as shown in the both the main manuscript and the ESI Section 1.8.3, provides further evidence on the well-defined crystalline Bragg reflections of CSA-LAG products. To clarify all these points we have added the paragraph below to the discussion. We think this clarifies any amorphization related queries to the reader. In addition, all the diffraction patterns in the revised ESI are now presented with the scattering contributions from the Kapton subtracted, providing clearer data for the reader. Thank you.

“Throughout all CSA-LAG experiments and PXRD analyses carried out in this work (147 diffraction experiments), we have observed no obvious evidence of amorphisation. Whilst amorphisation is possible by milling, the conditions of milling are critical. For example, neat milling can lead to amorphisation in some systems, whilst milling in the presence of solvent promotes the formation and growth of crystals in the mill^{27,33,41,56}. Since our method uses milling in the presence of solvent, we have seen no obvious evidence of amorphisation from the diffraction patterns. We note that the raw PXRD patterns contain ‘halo bumps’ due to the use of the Kapton tape to cover the sample (ESI section 1.8.1) and not amorphization, and that upon applying a Kapton background scattering correction, the resulting patterns have very well-defined diffraction peaks (as shown in Figure 4 and the ESI Figures S6 and S7). Whilst thermal analysis can help quantify the amorphisation, it cannot easily be used for our milled solids since they are wet which would lead to complex DSC thermographs containing many events including solvent evaporation and phase transitions of difficult resolution and interpretation. Further to this, for amorphisation to occur in the mill several conditions need to be satisfied: a) the solid forms under study must have a T_g above the milling temperature, b) high intensity milling is required, and c) low humidity must be ensured since water (even in small amounts) acts as a strong plasticiser. Given that our milling is performed at room temperature, is of relative low intensity and duration, and is carried out in the presence of solvents (with their consequential plasticising effects⁵⁷ and enabling crystal growth in the mill²⁷), amorphisation under our experimental conditions is extremely unlikely. Our observations are consistent with prior literature on the milling of hydrates⁵⁸ which state of the “impossibility to amorphise most hydrates by milling due to the plasticising effect of the structural water molecules which generally depresses the T_g of the corresponding amorphous forms below the milling temperature” -with similar expectations for solvates⁵⁷.”

5 - Considering that degradation can also occur during milling, did the authors check that no degradation occurred during ball milling?

Whilst degradations can occur at high milling intensities, our milling conditions are mild and thus degradation extremely unlikely for the systems studied, especially at our working temperature. From the 147 powder patterns involving this work and approximately 500 outcomes during the method development and further exploration; never showed any diffraction peaks that could not be explained from the pure phases, the solvates or the hydrates under study. So we did not observe degradation.

6 – Please review the reference list and ensure consistency in the use of journal names. Use either the full journal name or its standard abbreviation. We thank Reviewer 2 for spotting this issue with our bibliography, we have now reapplied uniform abbreviations across the main manuscript and our ESI.

=====

Reviewer #3 (Remarks to the Author):

The paper *Shaken, not Stirred!* is interesting for the fact that it provides a quick method to determine solvate forming propensities of APIs in the presence of solvents. It is also shown that depending on the solvent concentration, different solvates can be produced. All results obtained by grinding have been verified by crystallisations in slurries confirming all outcomes and demonstrating that grinding or mechanochemistry is the more rapid method.

There is one important problem in the paper. Solvent activities have been calculated from concentrations, however, the assumptions for the conversions are incorrect. Solvent activities in the mixture are calculated using the assumption of a mixture of two pure solvents and activities are obtained from the NRTL model based among others on vaporisation data of the solvent mixtures. If the solvents would have been the only constituents of the system, these activities are correct, however, APIs are present within the system too. Most of the API is present in the solid state, which could be interpreted in terms of an activity of 1, but the fact that hydrates or solvates are forming is absolute proof that the approximations of mostly pure solvent (all eqs. in Figure 1) is not valid as interactions between solvent and API are undeniably present and thus the activity of the solvent is affected by the presence of the API. Solvents in solvates possess a well-defined activity, which is equal to the vapour pressure of the solvent above the solvate. The authors are very aware that impurities affect solvent activity as they mention the importance to avoid the presence of water in the solvents, however, they ignore the presence of the API "impurity". Activities change in the presence of the APIs, I suggest therefore to use solvent concentrations and not activities to report the results, concentrations are undeniable.

The reason that slurries and grinding give rise to comparable results with similar "activities" is because the conditions under which the solvates form are very similar. In both cases the system in equilibrium will contain saturated solutions of API; thus, solvents with the exact same activities (in grinding and slurry experiments) tending towards equilibrium. Except, these activities are definitely not the ones calculated by NRTL assuming pure solvents.

We thank Reviewer 3 for this insightful comment.

Moving from concentrations to activities is one of the novelties of our work, and what makes the outcomes predictable from the presented thermodynamic framework! Our framework and controlling the activities rather than the concentrations, really consolidate outcomes across experiments all over the literature and adds significant advance, quality and predictive capacity. This is because the activities, and not the concentrations, are the ones driving the thermodynamics and the conversions. This is the key novelty and exceptional value of our contribution!

However, we agree that in the strict thermodynamic sense, the presence of the API in liquid can perturb solvent activities relative to those of the strictly pure binary solvent mixture. The solubilities of the APIs are very low and hence the deviation in the exact activity value for a solvent in the pure liquid versus the liquid with a small amount of API is very small. This is well known and the omission from the calculations (due to the huge complexity) is justified and standard practice! The equivalent practice was followed in the recent Nature publication regarding hydrates (<https://doi.org/10.1038/s41586-023-06587-3>). We have added the following paragraph to section 2.2 clarifying this:

“Importantly, our activity calculations are based solely on pure solvent-solvent equilibria, excluding the influence of dissolved API. This is a widely accepted approximation^{1,50} since it has been demonstrated that the presence of a small amount of API (due to the API’s solubility in the liquid mixture) only lowers the effective solvent activities by a few percentage points (e.g. -6% for theophylline⁵¹ and -4% for ampicillin¹⁵ for water activities in water:organic solvent mixtures). Given that solvent activities cannot be directly measured experimentally (except for water) and are usually estimated within the screening ranges attainable, this simplification introduces an error smaller than the typical experimental uncertainty and hence is justified.”

A related issue can be found in Figure 7. This figure appears to be or is even presented as a phase diagram, but it is not because one of the constituents, the API, is missing in the diagram. There should in fact be a fourth axis representing the API and one would expect the current drawing to represent an isopleth of the four-dimensional phase diagram. But even this is not the case, as in fact, the API concentration most likely changes with the saturation in the different solvent mixtures, so what we are looking at is most likely some curved surface in the quaternary system ($T = \sim 25^{\circ}\text{C}$). Therefore figure 7 is not a ternary solvent system, but a quaternary system in which the effect of the three solvents on the API under saturation (against hydrate/solvate/pure API..., surely all different API saturated concentrations) is demonstrated. It is far from a phase diagram even if this map may be of interest to crystal engineering. Thus this figure should be clearly described in what it represents and the term phase diagram should be avoided as this will cause confusion.

In most thermodynamic contexts, one could freeze one of the variables and deliver a phase diagram with a variable fixed. For example, solubility depends on temperature and pressure but the pressure dependence is always obviated (not explored) for practical reasons. Again, we could have the extra API constituent axis on Figure 7 but this will make the diagram impractical and very difficult to read, whilst the composition of the API in the liquid mixture will have a very small effect in the boundaries. Hence, it is not presented and, as discuss above, widely accepted as the gold standard in all industrial and academic practices.

Some minor issues

>The objective of the figure just below the abstract (TOC figure?) is not clear and how is the repetition of the grinding jar with and without laboratory background relevant? This figure does not capture in its present state the core of the paper.

Thank you for this suggestion. This has been improved now, we hope the reviewer likes the new TOC.

Page 9 top: refers to 4-hydroxybenzamide hemihydrate form I.

>chemical name is misspelled

Thank you for spotting out typo, it has now been corrected.

Section 2.6

>The entire discussion remains rather vague. What is meant with α (H₂O/DMF) \approx 5? is this the ratio between two activities? then write this explicitly with two alphas. Otherwise, what is it?

Yes, it is. The α ratio was defined earlier but we have now changed the nomenclature and use the actual ratio of the two alphas for clarification and as suggested by the reviewer. Thank you.

The discussion has been changed and extended to satisfy all referees. We hope this makes it stronger.

This discussion of should be clarified and indications as "very close" (4th line from the bottom of section 2.6) should be quantified. Activities run from 0 to 1, so differences of 0.1 are differences of 10% and are not necessarily 'close'. So be careful what is stated.

Thank you. We have changed this to "similar".

Figure caption 6, last line: Plots also show solid form transitions

>solid should be solid

Thank you that has been corrected.

Page 18, second paragraph, second line: it is very important to apply especial care in preparing

>especial – special

Thank you that has been corrected.

page 18, third paragraph, end: For scenario C, studying competing solvates work, we found that working with 5 μ L per mg of API solids (rather than 2 μ L per mg) helped reduce these uncertainties and led to observed activity ratios very close to those

predicted from the critical activity values of the independent equilibria.

>This sentence is not clear, what is very close? and how exactly are conditions predicted from the critical activity values. Prediction can be understood on many different levels. This refers to section 2.6? Please be more clear and explicit.

Thank you. This has been rephrased to:

“helped reduce these uncertainties and led to observed activity ratios **for solvate conversions of similar values** to those predicted from the critical activity values of the independent equilibria.”

Conclusions, first paragraph: By precisely controlling solvent activities, CSA-LAG enables the construction of complex phase diagrams across a range of solvation scenarios—including competitive solvation—with exceptional efficiency and accuracy.

>Conditions for the formation of solvates have been elucidated, but phase diagrams have not been produced or provided in this paper.

Here we disagree with the reviewer. The phase diagrams are defined by the boundaries that lead to phase changes, and the determination of the critical activities define the phase boundaries and hence enable the elucidation of a phase diagram.

Conclusions, second paragraph: This approach identifies solvent activity thresholds that govern solvate formation, ...

>The paper identifies solvent concentration thresholds, the activities are not properly defined

We disagree, please read the explanation above. We are adopting the community's best practice. The error by the omission of the API is very small (and smaller than the experimental uncertainty) and hence the approximation is widely used. The calculation of the activity with small amounts of API is very complex, in fact, not possible with current models. But the predicted values omitting the API presence are excellent.

>The caption of Figure F2 is not clear
This has been improved. Thank you.

Reviewers comments are provided in black, our response in blue and changes to the manuscript in red.

Reviewer #1 (Remarks to the Author):

The authors have satisfactorily addressed all the comments.

We sincerely thank reviewer 1 for their positive feedback and appreciation of our revisions. We are pleased that our improvements have addressed all previous concerns and we are grateful for their original discussion points raised and their supportive evaluation.

Reviewer #2 (Remarks to the Author):

The manuscript “Shaken, not stirred! Phase diagrams of Pharmaceutical Solvates”, by Cruz-Cabeza et. al., presents a novel and sustainable mechanochemistry-based method for screening the formation of solvates and hydrates of pharmaceutical compounds, utilizing the solvent activity of solvent mixtures. The authors have thoroughly addressed all of my previous comments and have significantly improved the revised version of the manuscript. The current version is well-written, and the main message is clearly conveyed, making it accessible to a broad scientific audience.

We thank Reviewer 2 for their encouraging feedback and are delighted that our revised and improved version conveys the message and significance of our work. We appreciate their recognition of the conceptual and methodological improvements in response to our previous version and are happy to hear that it is now accessible to a broad scientific audience.

Reviewer #3 (Remarks to the Author):

Unfortunately, the paper has not been sufficiently improved to justify publication. The authors state that moving from concentrations to activities is one of the novelties of their work; however, so far the authors have not convincingly demonstrated that using estimated solvent activities instead of solvent ratios provides an added value and leads to new insights.

1. As presented in the paper, the estimated solvent activities are nothing more than a more complicated way of representing the solvent ratios used.

2. The authors state that they can neglect the effect of the API on the solvent and that this is common practice citing two papers from the literature.

- The paper by Firaha et al. in fact does take into account API activity as they base their analysis on the measured water vapour pressure of hydrates. This vapour pressure is therefore clearly dependent on the interaction between water and the API. It does not support the approach of the authors.

- The paper by Khankari et al. states that water activity can be altered by changing its concentration in mixed solvents. This neither supports the tenet of the authors to use activity based on the vapour pressures of individual solvents in solvent mixtures while neglecting the presence of API.

3. The authors use liquid assisted grinding, which implies high local pressures with a massive excess of solid API in the presence of some liquid. Locally this may easily lead to high supersaturation, even if the system may relax to equilibrium shortly after. Saturation implies that the activity of the crystallising substance (the API or its solvate) is 1 in solution and for supersaturation the activity will be higher. Neglecting the effect of the API under these circumstances is not convincing from a thermodynamic point of view. It is the crystallising entity and therefore its activity is of importance.

4. Repeating what I stated in the original review: if a solvate is formed, there is necessarily an effect of the API on the solvent, the actual solvate formation (= interaction) proves this. Dilute solution approximations (solvent activity = 1) breaks down in such a case, because a solvate is not dilute.

The four points are addressed below.

Unfortunately, the paper has not been sufficiently improved to justify publication. The authors state that moving from concentrations to activities is one of the novelties of their work; however, so far the authors have not convincingly demonstrated that using estimated solvent activities instead of solvent ratios provides an added value and leads to new insights.

1. As presented in the paper, the estimated solvent activities are nothing more than a more complicated way of representing the solvent ratios used.

We respectfully disagree with Reviewer 3 on this point. The thermodynamics of real systems are governed by activities, not concentrations—a distinction that is foundational and non-negotiable within the discipline. These principles, established by Gibbs and others over a century ago, are well-documented and universally accepted. As such, they cannot be modified to suit alternative interpretations. This is clearly explained in standard undergraduate textbooks such as *Atkins' Physical Chemistry* and *Petrucci's General Chemistry*, as well as in widely accessible resources like the Wikipedia entry on thermodynamic activity. The opening sentence of that page states: “*Activity is a measure of the 'effective concentration' of a species in a mixture, in the sense that the species' chemical potential depends on the activity of a real solution in the same way that it would depend on concentration for an ideal solution.*” Since our work deals with real solutions, the use of activities is not only appropriate but essential.

Accurate prediction of outcomes in such systems requires the use of activities, and this is a point on which there is little room for scientific dispute. To reinforce this, we have added the following paragraph at the end of the Discussion section to clearly articulate the critical importance of working with activities:

“To the best of our knowledge, this study is the first to implement controlled activities in liquid-assisted grinding, representing a methodological innovation and a key strength of our approach. Using activities rather than concentrations is essential for capturing the non-ideality of solvent mixtures, ensuring transferability across different solvent systems, enabling meaningful comparisons between methodologies, and allowing prediction of solid form transformations under varying environmental conditions. For example, as shown in Table 2, the critical water activity for the NF-MH-I transition, measured using CSA-LAG in a h₂O:eg solvent mixture, is approximately 0.5—corresponding to a water mole fraction of 0.52 in that system. This activity threshold defines the thermodynamic limit for hydrate formation and remains consistent across solvent mixtures and experimental methods, thereby enabling robust prediction of behaviour. If NF is developed in its anhydrous form using a water:acetone mixture, maintaining water activity below 0.5 ensures the stability of the anhydrate; however, relying on mole fraction alone (e.g., 0.52) would be misleading, as this corresponds to a water activity of 0.76 in water:acetone, promoting hydrate formation. Furthermore, the critical activity determined via CSA-LAG can be extended to predict behaviour under liquid–vapour equilibrium conditions, such as relative humidity, whereas concentrations lack this transferability.”

2. The authors state that they can neglect the effect of the API on the solvent and that this is common practice citing two papers from the literature.

We do not claim that API-solvent interactions are absent or unimportant. We claim that **the dissolved API perturbs the bulk liquid phase solvent activity by amounts smaller than experimental uncertainty.** This is because solubilities of non-electrolytes are typically small and this has been empirically validated by direct measurements. Accounting for the API in the calculations is simply not possible because the coefficients do not exist because there is no data for this.

In the first revision of the paper we added citations to three papers and explained that the error is very small and certainly smaller than the experimental accuracy. We have extended the explanation further now including 6 citations and rephrased to the following:

“Importantly, these activity calculations consider only solvent-solvent equilibria, excluding the influence of dissolved API. This is a well-established simplification in pharmaceutical hydrate research for non-electrolyte systems, supported by decades of literature and experimental validation.^{16,51,52,1,53} At saturation, dissolved API concentrations are typically low and exert minimal influence on solvent activity — quantified at less than 6% in representative systems^{51,16} —well below our measured activity increments ($\Delta\alpha = 0.1$) and within typical VLE model uncertainties ($\pm 1-5\%$).⁵⁴ Industrial crystallisation guidance explicitly demonstrates the use of NRTL binary solvent activity calculations for controlling water activity during hydrate crystallisations, again without including API effects in the activity calculations⁵².”

There is a very large body of works using this approximation and we feel this small clarification of the paper should not need the addition of more references than the six already presented. The reviewer is further referred to the following literature works, all using this approximation:

- Firaha et al 2023 in reference 1, Nature paper
- <https://pubs.acs.org/doi/full/10.1021/acs.cgd.9b01066>
- [https://doi.org/10.1016/S0378-5173\(02\)00277-6](https://doi.org/10.1016/S0378-5173(02)00277-6)
- <https://pubs.rsc.org/en/content/articlehtml/2016/ce/c6ce01834c>
- <https://doi.org/10.1021/acs.cgd.1c01045>
- <https://doi.org/10.1039/C5CE01758K>
- <https://doi.org/10.1021/acs.cgd.7b00664>
- Also mentions nrtl <https://doi.org/10.1021/acs.cgd.6b01231>
- For a salt <https://doi.org/10.1021/acs.oprd.0c00260>
- <https://doi.org/10.1021/acs.molpharmaceut.5b00856>
- <https://doi.org/10.1021/acs.molpharmaceut.5b00357>
- <https://doi.org/10.1021/op7001497>

- The paper by Firaha et al. in fact does take into account API activity as they base their analysis on the measured water vapour pressure of hydrates. This vapour pressure is therefore clearly dependent on the interaction between water and the API. It does not support the approach of the authors.

With all the respect, we believe Reviewer 3 has actually stated our defence. As written in Firaha et al. in reference 1: “For all phase transition data, raw experimental results were used as much as possible, including the original data from **dynamic vapor sorption (DVS) experiments and slurry experiments.**”

Firaha et al **treat DVS and slurry data as equivalent and interactable sources of critical water activity values.** This is precisely our point.

- **DVS experiments** measure equilibrium water vapour pressure over solid hydrate(s) probing the solid vapor equilibrium.
- **Slurry experiments** use mixtures and calculations of activity from the literature just as we have done, without considering the minimal effect of the dissolved API on the activity.
- **Both methods probe the same thermodynamic quantity: the critical water activity.**

The fact that Firaha et al do not distinguish between these methods and actually accept the transformation boundaries from either approach provides further support for us and validates that the water activity is the fundamental thermodynamic variable regardless of whether it is measured by vapour pressure, calculated in solutions (slurries as we do) or controlled mechanochemistry (CSA-LAG, done for the first time in our paper). **Rather than contradicting our approach, Firaha et al, actually provide evidence that activity-based analysis successfully unifies diverse experimental methods that probe hydrate stability.**

- The paper by Khankari et al. states that water activity can be altered by changing its concentration in mixed solvents. This neither supports the tenet of the authors to use activity based on the vapour pressures of individual solvents in solvent mixtures while neglecting the presence of API.

Our apologies here, the Khankari paper was not relevant for this. The paper that was relevant here is “Reutzel-Edens, S. M., Braun, D. E. & Newman, A. W. Hygroscopicity and Hydrates in Pharmaceutical Solids. in *Polymorphism in the Pharmaceutical Industry* 159–188 (Wiley, 2018)” which is now reference 52 in the manuscript. The water activity is indeed changed by changing concentrations, but we do not understand why the reviewer is stating this. As explained in answer to point 1, activities are the important parameter.

3. The authors use liquid assisted grinding, which implies high local pressures with a massive excess of solid API in the presence of some liquid. Locally this may easily lead to high supersaturation, even if the system may relax to equilibrium shortly after. Saturation implies that the activity of the crystallising substance (the API or its solvate) is 1 in solution and for supersaturation the activity will be higher. Neglecting the effect of the API under these circumstances is not convincing from a thermodynamic point of view. It is the crystallising entity and therefore its activity is of importance.

No-body understands the specific mechanisms of mechanochemistry. For this paper, we do not need to know this because we only look at the thermodynamics and **thermodynamics outcomes are independent of pathways.**

Further to this, the reviewer is mistaken in the statement “Saturation implies that the activity of the crystallising substance (the API or its solvate) is 1 in solution and for supersaturation the activity will be higher.”. This is incorrect. Saturation means that the activity of the crystallising substance is the critical activity of that substance in the solution-solid equilibrium (the solubility!). The saturation activity of the API in solution however is never 1, it is in fact very close to 0 since solubilities of APIs are usually very low. An activity of 1 of the API in solution is only possible for the pure melted substance! **Activities can never be greater than 1.** Activities are 1 also in the solid state, but not in solution!

4. Repeating what I stated in the original review: if a solvate is formed, there is necessarily an effect of the API on the solvent, the actual solvate formation (= interaction) proves this. Dilute solution approximations (solvent activity = 1) breaks down in such a case, because a solvate is not dilute.\

We agree that the API will interact with the solvent but that does not contradict our approach. We do not understand what the reviewer means by “Dilute solution approximations (solvent activity = 1) breaks down in such a case, because a solvate is not dilute.\”.

The API interacts with the solvent, meaning that the API-active solvent is not ideal. However, the API solubility is very low, and there is a major carrier solvent also present which also interacts with the active solvent. So in the mixture we have the following interactions present:

- Active solvent – active solvent
- Active solvent – carrier solvent
- Carrier solvent – carrier solvent
- API – active solvent
- API – carrier solvent
- API – API

We do not say that API- active solvent or API – carrier solvent are negligible interactions. No. We do state, however, that the API is in significantly smaller fraction in the liquid than the other two solvent components, and hence, its obviation is justified since the effect of the API on the resulting active solvent activity is very very small, as shown experimentally.

Our conditions of work the mixed solvent is saturated with API, but the proportion of API in that solvent is very small. The statement “a solvate is not dilute” is confusing since the solution can be dilute a solvate is a solid phase so the concept of dilution is not really applicable in the solid state.

Finally, to give an example, the mol fraction of nitrofurantoin at 37 Celsius in pure water is 10^{-5} , so significantly smaller than the content of solvent molecules!

Reviewers' comments are in black, our responses in blue and changes to the manuscript in red

Reviewer #2 (Remarks to the Author):

After carefully reviewing the manuscript "Shaken, not Stirred! Phase Diagrams of Pharmaceutical Solvates from Mechanochemistry" by Cruz-Cabeza et al., alongside the comments from reviewer 3 and respective authors' detailed response, I conclude that two complementary perspectives on the thermodynamics of CSA-LAG experiments are presented. Reviewer 3 correctly notes that local supersaturation and transient non-equilibrium states during LAG may influence the nucleation kinetics of the APIs. While this observation is valid at the microscopic or mechanistic level, the approach presented by the authors remains thermodynamically sound for mapping solvate and hydrate equilibria. In the experiments presented by the authors, the API is present in large excess and has a very low solubility, such that its activity in solution is effectively constant and negligible. Consequently, the equilibria are governed primarily by the chemical potential of the solvent, which is appropriately described using experimentally determined activities for real, non-ideal mixtures, where activity coefficients deviate from unity and therefore the Raoult's law is no longer applied. Furthermore, the authors have provided empirical evidence confirming that the presence of dissolved API perturbs the solvent activity by an amount smaller than the experimental uncertainty. Therefore, the use of solvent activities, rather than simple concentration ratios, is a valid and necessary thermodynamic approach for defining solvate and hydrate formation boundaries. In my opinion, the concerns raised by reviewer 3 pertain mainly to local kinetic effects, which do not undermine the equilibrium-based rationale of the methodology presented by the authors. Overall, the authors have addressed all comments comprehensively and have further improved the clarity of the manuscript.

We sincerely thank Reviewer 2 for their thoughtful and balanced assessment of our work. We greatly appreciate their clear articulation of the thermodynamic principles underpinning our methodology and their recognition that the concerns raised by Reviewer 3 pertain to local kinetic effects which do not undermine the equilibrium-based rationale of CSA-LAG. We are grateful for their supportive engagement throughout the review process.

Reviewer #3 (Remarks to the Author):

The article on Controlled solvent activity liquid assisted grinding clearly improved since the first version; however, the authors do not sufficiently take into account a possibly breakdown of one of their main assumptions: 'that solubility is low and that therefore the effect of the API on the activity of the solvent can be neglected'. They state that this is common practice, while citing numerous papers.

First of all, I agree with the authors that activity (or the chemical potential) is the quantity of importance that controls phase behavior in the system. Where my point of view deviates from that of the authors is to what extent the presence of API and the LAG process itself may change the activity of the solvent and that therefore the activity listed by the authors is not necessarily correct. In physical chemistry textbooks such as the one written by Peter Atkins, it is clearly stated that the presence of a solute will affect the chemical potential of the liquid and vice-versa (p 140 Eight edition of Physical Chemistry by Peter Atkins and Julio de Paula). Moreover, the activity coefficient of the solvent changes as a function of the concentration of the solute and vice versa (Margulles equations)(p162 Eight edition of Physical Chemistry by Peter Atkins and Julio de Paula)

In dilute solutions, the presence of a solute may be neglected and I am aware that this approximation is being used in the pharmaceutical literature for slurries for example. When those assumptions break down will depend on the system under study and should in fact be tested. In particular, if one intends to provide accurate data based on activities.

My problem is that in this paper, the authors study liquid assisted grinding, while using approximations that have been accepted for different conditions, but not for LAG, because the combination of LAG and activities is new as the authors state.

In the paper it is stated that the liquid-solid ratio is 2 parts of liquid for 1 part of solid in terms of weight. This is not a dilute system. It implies that most solvent will be in contact with a solid surface, even more so during the grinding process. There will be an extremely large interface, while the calculated activity coefficients of the solvents are based on bulk mixtures with relatively little surface or interface. Even if the assumption that the solution is dilute due to low API solubility would be true, this is not necessarily sufficient; the system in LAG consists mainly of interface. The assumption that the presence of the (solid) solute can be neglected should be validated, which is not done in this paper.

Point i. Overall composition (ie 2 parts of liquid to 1 part of solid) in a heterogeneous system (a solid and a liquid) does not affect solubility in the liquid phase. Solubility is the concentration of the API in the liquid phase (ie 0.5g API/g solvent), which is independent of the amount of solids present since this is an invariant thermodynamic property.

Point ii. We appreciate the microscopic/mechanistic observation about interfacial vs. bulk activities. At steady state / thermodynamic equilibrium - which our 30-minute milling protocol ensures - the chemical potential (and therefore activity) of water must be uniform throughout the system, including at solid-liquid interfaces. Any gradient would drive mass transfer (accelerated by mechanochemistry) until equilibrium is restored. **The excellent agreement between CSA-LAG and slurry experiments ($R^2 = 0.975$, Figure 8) validates this assumption**, as slurry methods are established equilibrium techniques where bulk and interfacial activities are demonstrably equal. **We have shown correlation between both methods for 9 independent systems**, demonstrating broad transferability and generality.

Moreover, the authors state themselves:
p21, 122

"We encountered this issue for the 4ohbzm system in CSA-LAG with ace:eg. The high solubility of both the anhydrous and acetone solvate in the liquid mixture meant that CSA-LAG could only be performed by lowering the liquid to solids ratio to 1 $\mu\text{L}/\text{mg}$ which in turn led to a window of activities for which mixtures of the anhydrous form and the acetone solvate were observed."

The revised section now reads:

"Third, for instances where the API has a higher than usual solubility in both the active and the carrier solvent, the liquid to solid ratio needs to be reduced in order to avoid full dissolution. These conditions will lead to wider windows where the active solvent to API stoichiometries will lie below the stoichiometric reaction needs. We encountered this issue only for one of the systems: 4ohbzm in CSA-LAG with ace:eg. The unusually higher solubility of both the anhydrous and acetone solvate in the liquid mixture meant that CSA-LAG could only be effectively performed by lowering the liquid to solids ratio to 1 $\mu\text{L}/\text{mg}$ which in turn led to a window of activities for which mixtures of the anhydrous form and the acetone solvate were observed."

Here, the argument that the API exhibits low solubility is clearly not true.

We agree that higher concentrations may potentially lead to a larger error withing the "exact activity value". However, this -as argued before- still remains small (indicated by Grant et al., 1996 to be <10%). We need to emphasise that this situation would be encountered regardless of slurring or CSA-LAG. And that in 8 of the systems studied, this only occurred in one of them (the 4ohbzm system). The advantages of the method using activities are significant, even in the unusual scenario of higher solubilities (1 in 8) for which uncertainties may become slightly higher.

The authors state that using activities and linking those to the observed phase behavior (solvate formation) is much better than only relying on concentrations. I agree with the authors, but I disagree with the fact that the authors control the activity during the grinding process. What the authors have demonstrated in the paper is that their results coincide with those obtained by slurry under similar activities; however, how would the straight-line relationship look like in terms of liquid -liquid concentrations?

Liquid-liquid concentrations are just composition ratios and not transferable across solvent systems and don't provide any thermodynamic insight, whereas activities provide a thermodynamically rigorous, system-independent scale. This is why activities have been the standard metric in pharmaceutical hydrate research for over three decades. Please reflect further at Firaha 2022.

With respect to the second point "however, how would the straight-line relationship look like in terms of liquid -liquid concentrations?" Yes, activity is derived from concentration stating effective concentration. This would involve a single mathematical manipulation. Please refer to equations 1-4 in the ESI. As explained above, liquid-liquid mixtures do not provide either thermodynamic insight or transferable information.

The LAG process is much faster than slurries, which is absolutely of interest, but they have not demonstrated that within the LAG process, the critical solvent activity is the same for a given solvate while varying the solvent mixtures and keeping the activity of the active solvent the same.

The critical solvent activity at a specific temperature and pressure, must be the same for a given solvate in different solvent mixtures, or in the gas phase. This is proven by the thermodynamic equations and

laws, and must hold true due to the fundamentals of thermodynamics as the critical water activity is an intrinsic property. LAG activities behave just like slurry activities -as shown by our extensive comparison between both methods (a novelty of this work)- and the critical activities simply do not change with the inactive solvent used (ie ethanol, methanol, etc).

We appreciate Reviewer 3's desire for explicit demonstration of mixture-invariance. To address this directly, we have now added ESI Section 2.7 and ESI Tables S17-S18 showing that the NF-MH-I critical water activity (0.45-0.55) is reproduced using three different carrier solvents (H₂O:EG, H₂O:EtOH, H₂O:Acetone), despite these mixtures requiring dramatically different water mole fractions to achieve the same activity. This experimental validation confirms that phase boundaries are governed by activity, not composition, as expected by the thermodynamic laws that underpin solvation reactions.

We have also added an in-text mention of this: “Using activities rather than concentrations is essential for capturing the non-ideality of solvent mixtures, ensuring transferability across solvent mixtures **containing the same active solvent but a different carrier solvent (ESI, section 2.7)**, enabling meaningful comparisons between methodologies, and allowing prediction of solid form transformations under varying environmental conditions.”

Some of the statements in the paper are too strong if based on the current results and should be written in a less definite form.

p3, l21 The novelty of CSA-LAG lies on the use of solvent mixtures of precise compositions which provides the ability to precisely tune solvent activities in the milling environment.

Due to the grinding process, the extremely large interface, and in certain cases high solubility, "precisely" is not accurate here. It should be removed.

In the revised manuscript this now reads: “The central innovation of CSA-LAG lies in using solvent mixtures of defined composition to systematically and reproducibly tune the bulk solvent activity in the milling environment, enabling activity-based mapping of solvate/hydrate boundaries.”

This hopefully further clarifies the control of the bulk activities rather than local interfacial conditions.

p8, l14 This is a well-established simplification in pharmaceutical hydrate research for non-electrolyte systems, supported by decades of literature and experimental validation.

This may well be, but that doesn't mean that this simplification or assumption is valid for the current approach. It has not been convincingly established in this paper for this process.

We thank Reviewer 3 for this point. The revised manuscript now reads:

“This is a pragmatic approximation widely adopted across pharmaceutical crystallisation and hydrate studies for non-electrolyte systems⁵¹. Under these conditions, the bulk solvent activity (from validated VLE models) serves as an adequate control parameter. Consistent with prior hydrate literature¹, we adopt the same pragmatic approximation for CSA-LAG, while explicitly acknowledging potential limitations. Experimental validation^{16,52}, has shown, at saturation, dissolved API concentrations are typically low and exert minimal influence on solvent activity —quantified at less than 6% in representative systems^{16,52}—well below our measured activity increments ($\Delta\alpha = 0.1$) and within typical VLE model uncertainties ($\pm 1-5\%$)⁵³.”

Further to that, the confirmation of the transferability of this common simplification from solution crystallisation to milling is confirmed by the excellent R^2 of 0.975.

p8, 119 Industrial crystallisation guidance explicitly demonstrates the use of NRTL binary solvent activity calculations for controlling water activity during hydrate crystallisations, again without including API effects in the activity calculations.

Indeed, however, the guidance does not include new approaches such as LAG. The working hypothesis should be tested.

We thank Reviewer 3 for helping us strengthen and improve this point.

In response to “The working hypothesis should be tested”, please refer to the comparative correlation between solution and LAG results of $R^2 = 0.975$ in Figure 8.

In the revised manuscript this now reads:

“Industrial crystallisation protocols routinely rely on NRTL-derived bulk water activities to control an/hydrate equilibria in solution crystallisation. While this guidance does not explicitly address mechanochemical (LAG) environments, our results indicate that the bulk solvent activity remains an informative and historically overlooked control variable for CSA-LAG when the API is present predominantly as a solid and its dissolved fraction is small. We therefore use NRTL-calculated activities as a practical activity scale here, while explicitly delineating the limits of this approximation and outlining future validation experiments in Section 2.8⁵¹.”

p8, l23 While water activities can be measured directly (e.g., using a calibrated hygrometer), activities of other solvents cannot be experimentally determined.

This is not true and should be removed. Vapor pressures can be measured and they are a direct representation of the Gibbs energy, otherwise the NRTL calculations carried out by the authors would not be possible either.

We agree with Reviewer 3 that the vapor pressures can be measured and then relying on the partial vapour pressures (ie Antoine equation) and the modified Raoult's law, all activities can be calculated. Our original sentence was imprecise. What we intended to convey is that unlike water (for which RH sensors provide a ubiquitous direct readout of activity), there is no widely used "plug and play" hygrometer analogue for organic solvents; with organic solvent activities typically calculated from measured pressures, headspace GC, IR, MS, dew point methods, etc. with calibration and temperature control.

The revised and improved section now reads: "CSA-LAG enables activity-controlled mapping of solvation landscapes beyond water. Water activity can be probed directly (e.g., calibrated RH meter), whereas non-aqueous solvent activities are experimentally determined from vapour-pressure/VLE measurements or headspace analysis; here we use NRTL derived bulk activities from curated VLE data as a practical, transferable scale."

p8, l24 As a result, the thermodynamic solvation behaviour of non-aqueous solvents has historically been overlooked, with solvates often discovered serendipitously rather than through systematic investigation.

As the previous statement is incorrect, it cannot be the reason for the serendipitous discovery of solvates. It is more likely that the interest is less, because solvates in APIs are something to avoid, not to aim for. Water is so relevant, because it is all present and acts often as a solvent for the API in the body. The sentence is misleading at the least and had better be removed.

We agree that our phrasing was too strong. Non-aqueous solvates are not ignored; rather, hydrates have historically received disproportionate emphasis due to their ubiquity, pharmacological relevance, and regulatory impact. Whilst standardised, activity-based mapping across other solvents appears almost non-existent in the current literature. We have revised the manuscript framing our contribution as providing a unified, activity-controlled protocol for diverse solvents.

The revised section now reads: "Because hydrates have historically dominated crystallisation form screenings owing to their ubiquity and regulatory relevance, activity-based mapping of non-aqueous solvates is less common. CSA-LAG provides a unified, reproducible protocol to determine solvation boundaries across diverse solvent systems."

p22,

11

For example, as shown in Table 2, the critical water activity for the NF-MH-I transition, measured using CSA-LAG in a h₂O:eg solvent mixture, is approximately 0.5—corresponding to a water mole fraction of 0.52 in that system. This activity threshold defines the thermodynamic limit for hydrate formation and remains consistent across solvent mixtures and experimental methods, thereby enabling robust prediction of behaviour. If nf is developed in its anhydrous form using a h₂O:ace mixture, maintaining water activity below 0.5 ensures the stability of the anhydrate; however, relying on mole fraction alone (e.g., 0.52) would be misleading, as this corresponds to a water activity of 0.76 in h₂O:ace, promoting hydrate formation.

I agree with the reasoning here, but it lacks proof. Why did the authors not show three different solvent

mixtures in which the critical water activity is 'calculated' to be the same. Right now, the paper demonstrates that LAG is very efficient in creating solvates, but not that the activity is controlled.

Thermodynamic activities dominate and control equilibria independently of observing species present. As such activities are transferable across solvent mixtures. We appreciate Reviewer 3's desire for explicit demonstration of mixture-invariance. To address this directly, we have now added ESI Section 2.7 and ESI Tables S17-S18 showing that the NF-MH-I critical water activity (0.45-0.55) is reproduced using three different carrier solvents (H₂O:EG, H₂O:EtOH, H₂O:Acetone), despite these mixtures requiring dramatically different water mole fractions to achieve the same activity. This experimental validation confirms that phase boundaries are governed by activity, not composition, as expected by the thermodynamic laws that underpin solvation reactions.

We have adapted the section the reviewer is referring to, to refer to one of the specific examples shown in the ESI.

Thus within experimental uncertainty, the observed critical water activities are mixture independent, (something already known and accepted by the community), while the mole fractions at the boundaries vary strongly with the carrier solvent selection—precisely the reason activities, not compositions must be used. We respectfully clarify that critical solvent activities are intensive thermodynamic properties that, by definition, must be mixture-invariant at constant temperature and pressure. We have now provided explicit experimental validation of this principle for the NF system (ESI Section 2.7).

The authors use an approximation. It is their task to demonstrate that the approximation is valid within their system. It is not enough to state that it is common practice under other experimental settings. The authors should mention the shortcoming that the premise of controlled activities within LAG has not yet been fully demonstrated. If this is mentioned in the paper and the textual changes mentioned above have been implemented, this paper can be published.

Our entire paper demonstrates that CSA-LAG using activities of solvents determined in the bulk leads to identical results to well established slurry experiments (currently the golden standard for critical water activity estimation).

Included in this publication are 67 slurry experiments, 163 CSA-LAG experiments across 9 varying systems and extensive comparison of all hydrate literature, excluding the early work on developing these methodologies. And they all agree in all cases, demonstrating the generality of this work. This is as expected, since activities are transferable across solvent mixtures and scales, as you would expect from the thermodynamic foundations.

We have done this for 9 systems in this paper. And therefore, demonstrate the robustness of CSA-LAG. **CSA-LAG is a robust methodology that achieves the same thermodynamic endpoints as slurry experiments, but significantly faster.** We have demonstrated with extensive experimentation comparing our method to the prior one, the validity and robustness of our new methodology.

We have further added a section on hydrate understanding limitations, considerations and directions moving forwards in the discussion section which now reads as:

“Regarding scope and future directions, CSA-LAG maps solvation boundaries by varying bulk solvent activities obtained from VLE data, while the API remains largely a solid. This pragmatic assumption for non-electrolytes has been standardised across the crystallisation field over the last three decades for

hydrate/solvate equilibria^{1,15,16,18,51,52,54} and agrees with independent slurry equilibria across our set ($R^2 = 0.975$, Figure 8), indicating that the endpoints reached by LAG are thermodynamically meaningful. We recognise that LAG milling can lead to high interfacial area and localised transient supersaturation, where system-dependent short lived deviations from the bulk activity can be expected. Our analysis, however, relies on the equilibrated solids post milling rather than on the transient pathway or intermediates.

Whilst we acknowledge that high dissolved fractions can effectively perturb the bulk activity in comparison with the predicted NRTL activities, this represents an opportunity for methodological refinement rather than a limitation. For most pharmaceutical systems, low-moderate solubilities ensure negligible perturbation of solvent activities, well within typical VLE model uncertainties. In exceptional cases of higher solubilities (eg., 4OHBZM-acetone), we employ practical mitigation strategies: reduction of liquid:solid ratio, or selection of passive solvents that suppress dissolution. Where high solubility is unavoidable, NRTL derived activities still remain informative predictors of phase behaviour, with appropriate wider uncertainty bounds.

Moving forward, several opportunities exist to further enhance the methodology and the general understanding of hydrates. First, direct headspace activity measurements would provide explicit validation and understanding to quantify and potentially model solubility induced perturbations. Second, temperature-controlled milling using jacketed grinding jars would enable systematic mapping of critical activities as a function of temperature, decoupling thermal from compositional effects and expanding CSA-LAG use for full thermodynamic phase diagram construction. Third, time resolved in situ synchrotron diffraction experiments under CSA-LAG will further demonstrate the efficiency of our method and shed light on the transformation pathways. These future studies, which are ongoing, would build upon CSA-LAG's demonstrated ability to rapidly and reproducibly map solvate landscapes, transforming hydrate screening from a month-long trial and error to a single day thermodynamically guided approach."